# Minimisation of metabolic networks defines a new functional class of genes

Giorgio Jansen [1,2], Tanda Qi [3], Vito Latora [4,5], Grigoris D. Amoutzias [6], Daniela Delneri [3], Stephen G. Oliver [1,7] ✉ & Giuseppe Nicosia [1,2,7] ✉

Construction of minimal metabolic networks (*MMN*s) contributes both to our understanding of the origins of metabolism and to the efficiency of biotechnological processes by preventing the diversion of flux away from product formation. We have designed *MMN*s using a novel in silico synthetic biology pipeline that removes genes encoding enzymes and transporters from genome-scale metabolic models. The resulting minimal gene-set still ensures both viability and high growth rates. The composition of these *MMN*s has defined a new functional class of genes termed Network Efficiency Determinants (*NED*s). These genes, whilst not essential, are very rarely eliminated in constructing an *MMN*, suggesting that it is difficult for metabolism to be re-routed to obviate the need for such genes. Moreover, the removal of *NED* genes from an *MMN* significantly reduces its global efficiency. Bioinformatic analyses of the *NED* genes have revealed that not only do these genes have more genetic interactions than the bulk of metabolic genes but their protein products also show more protein-protein interactions. In yeast, *NED* genes are predominantly single-copy and are highly conserved across evolutionarily distant organisms. These features confirm the importance of the *NED* genes to the metabolic network, including why they are so rarely excluded during minimisation.

Two developments place Synthetic Biology at the threshold of a golden era. First, the wide application of the CRISPR-Cas technology enables wholesale genome engineering with almost any biotechnological organism[1]. Second, designer genomes, such as Yeast 2.0, facilitate pathway engineering and chromosome reconfiguration[2]. Now, it is necessary to strip down the metabolic network of key organisms such that they provide a basic workbench on which novel pathways may be elaborated while minimising diversion of metabolites away from the desired product[3,4]. To enable construction of such workbenches, we have developed a computational pipeline that can define a minimal metabolic network for a broad range of organisms. We exemplify the

pipeline's utility using *Saccharomyces cerevisiae* and demonstrate significant quantitative and qualitative differences between the minimal networks required for fermentative and respiratory growth.

## Results and discussion

The pipeline starts with a stoichiometric model of an organism's metabolic network and the inventory of genes that are required to encode all the enzymes and transporters that catalyse the reactions contained in that model. Thus the genome of the starting (wild-type) strain contains all these genes and our algorithm seeks to minimise the number of active reactions (i.e. maximise the number of genes

[1]Department of Biochemistry, University of Cambridge, Cambridge, UK. [2]Department of Biomedical & Biotechnological Sciences, University of Catania, Catania, Italy. [3]Manchester Institute of Biotechnology, University of Manchester, Manchester, UK. [4]School of Mathematical Sciences, Queen Mary University of London, London, UK. [5]Department of Physics and I.N.F.N., University of Catania, Catania, Italy. [6]Bioinformatics Laboratory, Department of Biochemistry & Biotechnology, University of Thessaly, Thessaly, Greece. [7]These authors contributed equally: Stephen G. Oliver, Giuseppe Nicosia. ✉e-mail: sgo24@cam.ac.uk; giuseppe.nicosia@unict.it

removed from the initial inventory) while keeping the biomass formation rate predicted by *Flux Balance Analysis*[5] above a strict threshold based on the wild-type value. The algorithm then selects a *"minimal" metabolic network (MMN)*, as a mutant strain in which all the reactions that are still active are "essential", i.e. no more genes encoding enzymes or transporters can be deleted without violating the growth rate threshold, which was set to a reduction of either 1% or 10% of the wild-type value[6] (Supplementary Fig. 1). To further improve the reliability of the results, we added another constraint, namely that any gene in the model that was annotated in the literature as essential[7] (Saccharomyces Genome Database – SGD; https://www.yeastgenome.org) could not be deleted by our algorithm.

We first focused on the metabolic network of *S. cerevisiae* s288c, using the consensus yeast genome-scale stoichiometric model, version yeast 8.3.1[8] (see Methods and Supplementary Table 1). Environmental conditions, and particularly the chemical composition of the growth medium, affect not only the metabolic behaviour of the cell, but also determine whether a given gene or reaction is essential for growth[9,10] and, thus, the definition of the minimal genome itself[11]. It follows that different *MMN*s will be predicted in different external conditions. In the genome-scale models, these are defined as a set of exchange reactions whose lower bounds regulate the maximum possible uptake rates of nutrient compounds. In our study, we have considered two commonly used growth media for *S. cerevisiae* as well as a defined minimal medium generated by an ad hoc algorithm (see Methods). For all three media, simulations were performed under both aerobic and anaerobic conditions. In the following, if not otherwise specified, we always include one of the most complex chemically defined media (SD), in both aerobic and anaerobic conditions, to enable comparison with previous studies[12,13].

Given these initial settings of the constrained optimisation problem, our pipeline employs an evolutionary algorithm[14], developed from a greedy hill-climbing approach[15], to iteratively select the reactions/genes to be deleted in order to find the *MMN*s. Over a maximum number of generations, chosen as a trade-off between the computational cost and the ability to explore different minimisation strategies, the algorithm selects new gene deletions for a given set of strains that constitutes the current population of candidate solutions. It promotes those solutions that have a greater Hamming distance (a metric for measuring the edit distance between two strings of characters) from the others. The aim of this approach is to maximise the diversity of strains in the current population. When the number of deletion attempts grows, and a candidate solution is not improved (i.e. no new deletions are selected), it is increasingly probable that this solution will be discarded and substituted by an "ancestor" solution obtained by backtracking the corresponding leaf of the search tree so that other branches can be explored (see Methods for a complete description of the algorithm). When the algorithm reaches the last generation, a post-processing procedure is launched to expunge the solutions that show no improvements and extract the genuine, and distinct, *MMN*s. This procedure also performs a series of further analyses, including the evaluation of flux distributions using *parsimonious Flux Balance Analysis* (see Methods). The number of unique *MMN*s found is between 750–800 for all six conditions studied (see Supplementary Table 1) with three simulations being performed for each condition.

In a parallel study, we examined whether our pipeline could predict which genes encoding enzymes or transporters used in the metabolic model were essential. The commonly used definition of essentiality for *S. cerevisiae* is that deletion of an essential gene results in failure to grow on a yeast extract/peptone/glucose (YPD) agar plate. These are extremely permissive conditions in which, for instance, auxotrophic or respiratory-deficient deletants are still able to grow. This exercise is complicated by the fact that the deletion of certain genes will restore viability to strains carrying a deletion in an essential gene. This phenomenon has been termed 'synthetic rescue' by Motter

et al.[16] and 'bypass suppression' by van Leeuwen et al.[17]. The latter authors performed a genetic screen to reveal cases of bypass suppression and thus distinguished between 'dispensable' (able to be suppressed by bypass mutations) and 'core' essential genes (unable to be suppressed). Amongst the cases of bypass suppression, they were able to identify those due to loss-of-function (LoF) mutations (equivalent to our deletion mutations). We found that 36/45 indispensable or 'core' essential genes defined by the in vivo experiments of van Leeuwen et al.[17] were never lost from our *MMN*s. It should be noted that the genetic screen of van Leeuwen et al.[17] was an initial survey whose size was insufficient to test all possible query genes in their assay, and therefore it was unable to identify all possible LoF bypass suppressors (C. Boone, personal communication). Details of these analyses may be found in Supplementary Data 4.

Figure 1 shows the basic properties of the *MMN*s in the six different conditions and with a maximum of 1% reduction in the rate of biomass formation. The Figure shows that the most significant difference between all the *MMN* solutions for *S. cerevisiae* is between the *MMN*s found under the three anaerobic conditions studied and those arrived at for aerobiosis. The minimal solutions in the anaerobic states are significantly smaller in terms of the number of genes involved. The other notable difference in the distributions is between the two richest growth conditions (YPD and SD media) and the minimal medium. The *MMN*s found by the algorithm for the two complex media contain fewer genes than those generated for the minimal nutrient condition; this phenomenon is even more evident when comparing the anaerobic conditions. Clearly, the absence of complex biochemicals (e.g. amino acids and nucleic acid bases) in the simulated minimal external environment forces the algorithm to retain more genes involved in biosynthetic pathways. When we compared the genes present in the *MMN*s in the different conditions, we found that relatively more genes (by *ca.* 2–5%) involved in transport functions (including amino acid transport) are retained in the *MMN*s for growth in the SD medium (see Supplementary Figs. 2–3; a full list of genes encoding transporters and their biological roles is given in Supplementary Data 5). This may be explained by a need to import a larger number of external complex metabolites into the cell and its compartments. In agreement with this, the aerobic and anaerobic versions of the *MMN*s for SD medium retain a similar number of transporter genes (Supplementary Table 6).

After considering the global properties of the metabolic networks, we next performed an extended analysis on the genes of the *MMN*. Supplementary Table 1 summarises the genes/reactions present in the *MMN*s, the frequency with which genes/reactions appear in them, the genes that are never present, and the essential genes excluded from our minimisation pipeline. The same analysis was performed for all 6 conditions examined as well as for sub-sets of the conditions, e.g. the three aerobic conditions or the three anaerobic conditions. A striking feature revealed by these analyses is that there is a set of genes that are present in >95% of all *MMN*s when using the 1% threshold for a reduction in growth rate. We call these the *NED genes* and use the term *NED degree* to denote the fraction of *MMN*s in which a given *NED* gene is retained. We find the number of genes that are *NED*s in the *MMN*s generated for growth in the minimal medium is larger than in all other conditions and is also larger for aerobiosis than anaerobiosis. (Full details on the number and properties of *NED* genes for the different growth conditions may be found in Supplementary Tables 1–5). For the *S. cerevisiae* genome-scale metabolic model, we find that there are seven genes (the 'Magnificent Seven') that are present in all the *MMN*s found by our pipeline in all the conditions: *TPS1, TPS2, CHO1, ADE3, YNK1, GPT2, PFK2*. Of these seven, three (*TPS1, TPS2, PFK2*) encode components of multiprotein complexes, four (*CHO1, ADE3, YNK1, GPT2*) encode enzymes that participate in multiple pathways, and two (*ADE3, GPT2*) catalyse multiple reactions[7,18]. All of these genes, with the exception of *ADE3* (and possibly *CHO1*), have a high

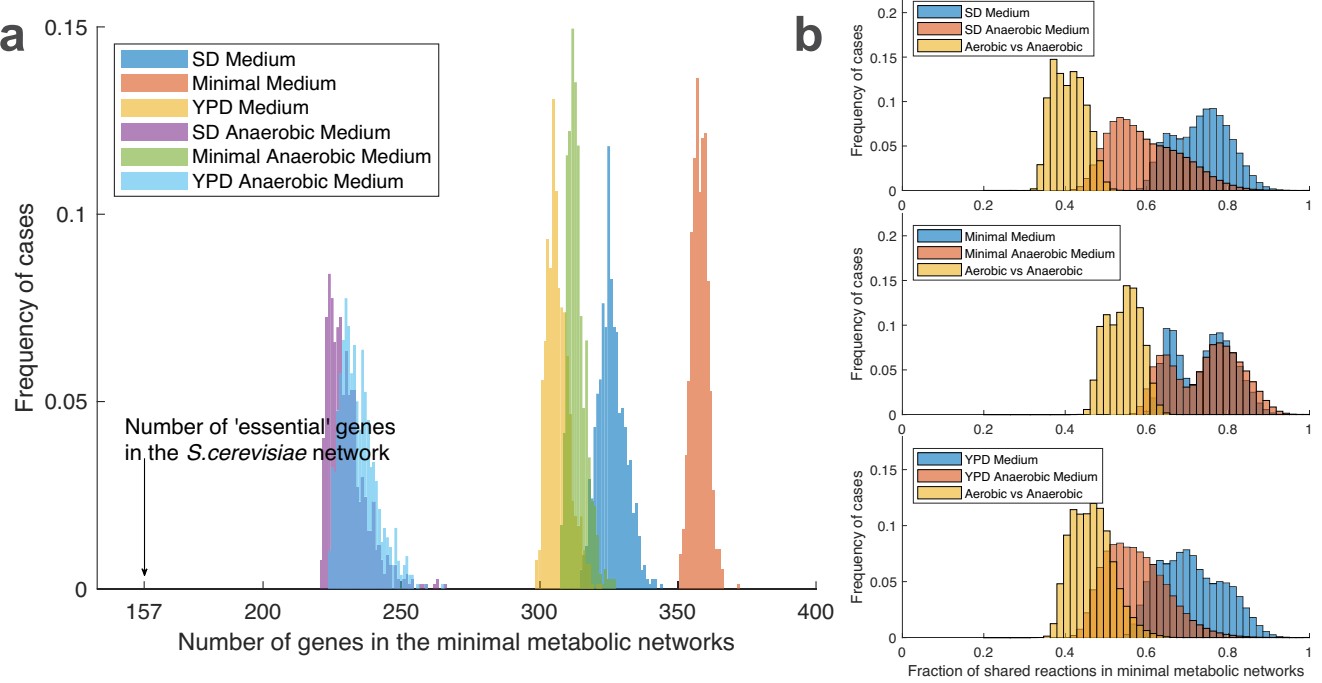

**Fig. 1 | Basic properties of the S. cerevisiae MMNs under 6 different growth conditions.** A growth-rate penalty of no more than 1% was employed. **a** Frequency Distributions of active genes in the *MMN*s; different clusters are present for aerobic and anaerobic conditions. In minimal media, both in aerobiosis and anaerobiosis, larger minimal metabolic networks are required than in richer media. **b** Fraction of *MMN*s sharing active reactions in a pairwise comparison in the corresponding aerobic and anaerobic conditions. Considering networks in the same conditions, the fraction of shared reactions is 60 - 90%, while comparing networks in different conditions, the fraction is 30 - 50%. Source data are provided as a Source Data file: Fig. 1 Source Data.xlsx.

number of experimentally determined genetic interactions[19]. These characteristics of multiple functions and functional interactions would make the absence of any of the 'Magnificent Seven' genes extremely difficult for the metabolic network to circumvent.

Our initial analysis of the Magnificent Seven provides a qualitative understanding of their importance for the metabolic network. We first considered different metabolic networks resulting from their deletion and measured the reduced capability to produce precursors for the biomass pseudo-reaction, as defined in the metabolic network, by *FVA* (Flux Variability Analysis; for details, see **FBA, FVA, pFBA** in Methods and detailed results in Supplementary Table 2 and Supplementary Data 1–2). However, it was essential to generate a quantitative measure of the impact of these and, indeed, all *NED* genes to the metabolic network beyond the *FBA* simulations and to consider whether they had a wider role which justifies their designation as a new functional class of genes. To achieve this, we first ran a Monte-Carlo simulation that employed two different strategies: the first involved removing seven randomly chosen non-*NED* genes (for either aerobic or anaerobic conditions) from the network; while, in the second, seven randomly chosen *NED* genes (again for either aerobiosis or anaerobiosis) were removed. In a final analysis, the Magnificent Seven *NED* genes (which are *NED*s in both aerobic and anaerobic conditions) were removed. To assess the network structural connectivity, we have evaluated the network global efficiency, which quantifies how efficiently the nodes of a network can exchange a signal, assuming communication takes place over shortest paths[20] (see Methods). Figure 2 shows the results of these simulations when the global efficiency of the yeast metabolic network was determined under either aerobic (Fig. 2b) or anaerobic (Fig. 2c) conditions. The results show that the removal of seven non-*NED* genes has little effect on the global efficiency of the network. In contrast, the removal of seven *NED* genes, and particularly the deletion of all the Magnificent Seven genes, severely compromises the efficiency of the metabolic network.

The metabolic network is a *functional* interaction network in which the nodes are nutrients and metabolites, and the edges are the enzyme-catalysed reactions that either transport or transform those nutrients and metabolites. In contrast, many of the other large-scale networks studied in functional genomics (most notably protein-protein interaction network[18,21]) are *physical* interaction networks. Genetic interaction networks, however, are *functional* interaction networks[3] studied by determining the functional, or epistatic, interactions between mutations (usually deletions) between gene pairs[19]. Indeed, an analysis of these epistatic relationships in yeast has been used to validate and correct the interactions built into the genome-scale metabolic model of *S. cerevisiae*[22]. For these reasons, it was important to establish whether the *NED* genes played a pivotal role in yeast's genetic interaction network. This was done using a parallel set of Monte Carlo simulations using a genetic interaction sub-network comprising only those genes that encode the transporters and enzymes involved in the metabolic network. The only difference being that no data are shown for the impact of the deletion of either non-*NED* or *NED* genes on the genetic interaction network because these interactions have only been determined in aerobiosis. Again, we see that the deletion of a random set of *NED* genes has a much more profound effect on global network efficiency than does the equivalent deletion of non-*NED* genes (Fig. 2a). Interestingly, the effect of deleting all members of the Magnificent Seven is even more marked in this case, perhaps indicating that these seven genes have additional biological roles, outside the metabolic network.

Finally, in order to investigate the general biological context of the *NED* genes, we compared three important characteristics of these genes and their protein products with those of both the essential and non-*NED* genes found in the wild-type metabolic network of *S. cerevisiae*. All three of these characteristics are of critical importance to a functional metabolic network: gene-gene interactions, protein-protein interactions, and paralogy. By examining the protein-protein

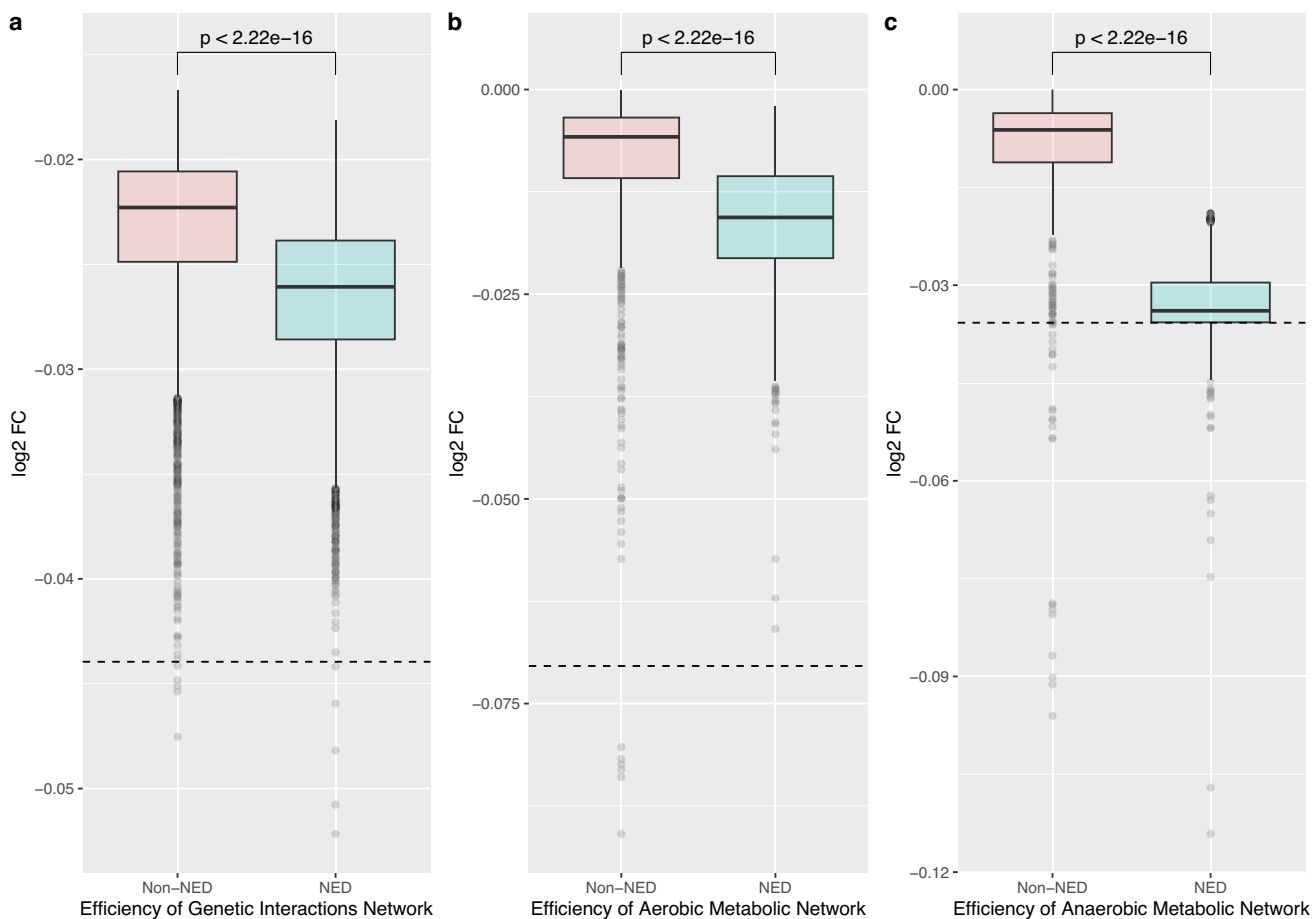

**Fig. 2 | Efficiency of metabolic networks, in aerobic (Fig. 2b) and anaerobic conditions (Fig. 2c), and of the genetic interaction network (under aerobic conditions only) upon deletion of seven *NED* or non-*NED* genes (Fig. 2a).** The horizontal dashed lines correspond to the efficiency of the networks when all of the magnificent seven genes have been removed. Bounds of boxes defined as Q1 - Q3, i.e. 25th − 75th percentile; bounds of whiskers defined as Q1-1.5xIQR - Q3 + 1.5xIQR. A Mann-Whitney two-sided test was used for statistical significance. All *p*-values are below 2e-16. A full account of the statistical basis of these box plots is given in Supplementary Data 8. (Supplementary Figs. 13–15 provide the results of classical network topology analyses, e.g. average degree and weight distributions. Readers may visualise the impact of minimisation on network topology by referring to the metabolic maps, Supplementary Figs. 8–12, a link to the files that will enable readers to generate their own maps is also provided). Source data are provided as four Source Data files: Fig. 2 Statistical Data Boxplot.docx, Fig. 2a Source Data.csv, Fig. 2b Source Data.csv and Fig. 2c Source Data.csv.

interactions network[23], we found that (on average) the products of *NED* genes have a significantly higher number of interactions (on average, 2.4 PPIs) than those of non-*NED* genes (on average, 0.8 PPIs), and even than those of the essential genes (on average, 1.1 PPIs); all the above differences are statistically significant (Mann-Whitney test, *p*-values < 0.05). Reassuringly, in the genetic interactions network[19], the *NED* genes also have more interactions than the non-*NED* genes (on average, 348 vs 238 GIs; Mann-Whitney test *p*-value < 0.05), although the essential genes are the most connected overall (on average: 497 GIs; see Fig. 3). Furthermore, the *NED* genes have a significantlty higher fraction of singletons (genes with no paralogs)[24] than the non-*NED* genes (69% vs 32%; Chi-square test *p*-value < 0.05).

It is important to point out that our results and analyses are based on the genome-scale metabolic model and, particularly, on the definitions of external conditions and biomass composition. Definitions of the biomass function are based on very old biochemical analyses that used techniques that lacked the quantitative and qualitative accuracy of modern mass spectrometric techniques, thus updates of the biomass composition could have a profound impact on the results[25]. The *MMN*s we found are then a minimisation of the specific metabolic network and its related genes, aimed at maintaining the cell's activity as defined for the wild-type organism, rather than a minimisation of

the overall number of genes for a universal biomass definition. Such a comparative task might be undertaken in the future and would be of interest from both a biological and a computational point of view. Thus, optimal pathways could be excised from a number of different species to be included in a synthetic minimal cell. However, this would require the definition of a global "life" function. Such a function could be inferred by the studies on minimal organisms, such as the JCVI-syn3.0[26]. We have made a start on this task by constructing *MMN*s for a range of species from bacteria to mammals. These ranged from *Mycoplasma genitalium* (a wall-less bacterium with a very small genome) to *Homo sapiens*. First, we minimised the metabolic networks of three other yeast species in the Order Saccharomycetales: *Schizosaccharomyces pombe*, *Komagataella phaffii* (syn. *Pichia pastoris*) and *Yarrowia lipolytica*. For three of these yeast species (*S. cerevisiae, Sz. pombe,* and *K. phaffii*), it was possible to compare results on the same minimal medium and to exploit experimentally determined data on gene essentiality. To appreciate the power of these comparisons between yeast species, it should be noted that the evolutionary distance between *S. cerevisiae* and *Y. lipolytica* is as great as that between *H. sapiens* and a primitive chordate, the sea squirt *Ciona*[27].

Analysis of the metabolic models constructed for multicellular organisms, such as *H. sapiens* or *M. musculus* presents particular

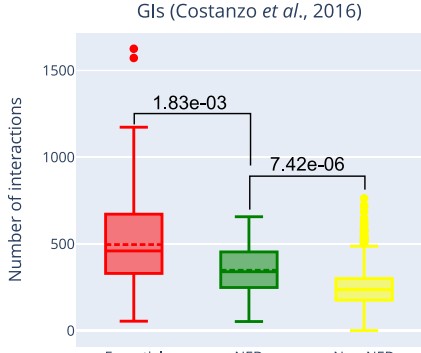
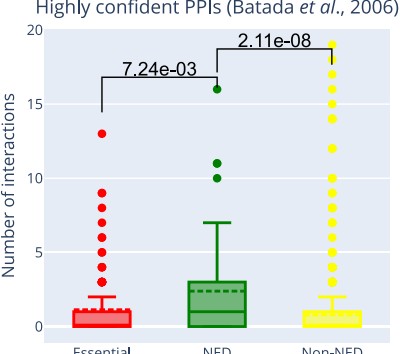
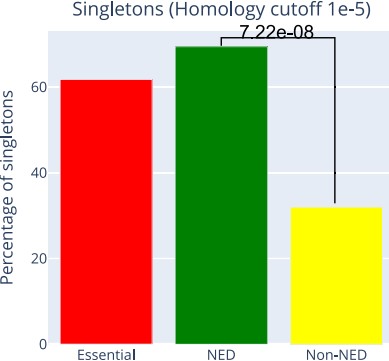

**Fig. 3 | Graphical summary of the properties of *NED* genes (*N* = 49) compared to non-*NED* (*N* = 925) and essential genes (*N* = 157) in *S. cerevisiae*, considering aerobic conditions.** The *NED* genes have more protein-protein and genetic interactions than the non-*NED* ones. The protein-protein interactions (ppi) for these genes are even higher than for the essential genes, while the essentials have more genetic interactions than all the other genes on average (Mann–Whitney two-sided test used for statistical significance). Moreover, a significantly higher fraction (69%) of *NED* genes are singletons (do not have a paralog) compared to non-*NED* genes (32%; Chi-square test *p*-value < 7.2e-8). Thus the *NED* genes rarely have back-up copies, making them very difficult to remove during the construction of the *MMN*s.

Genetic interactions were based on Costanzo et al. [19], protein-protein interactions were based on the highly confident interactions set of Batada et al. [23]. Singletons were calculated with BLASTp and homology cut-off of E-value < 1e-5. For the box-plots, each box spans from the first quartile (Q1) to the third quartile (Q3). The horizontal line inside the box represents the center/median value (Q2), while the dotted horizontal line represents the mean. The box whiskers span from the box's edges +/− 1.5 times the interquartile range (IQR: Q3-Q1). The dots outside the box whiskers represent the outliers. Source data are provided as a Source Data file: Fig. 3 Source Data Source File.xlsx.

problems, both because it is difficult to specify the inputs to and outputs from the metabolic networks, but especially because different types of cells, tissues, and organs have different metabolic responsibilities with their own inputs and outputs. Thus, while the Recon3D model of human metabolism despite being the largest of all the models we considered (requiring 2248 genes to encode its enzymes and transporters) led to very small *MMN*s (with 134 genes, on average), the Chinese Hamster Ovary (CHO) cell line produced an *MMN* containing 170 of the 1766 of the genes required for the wild-type model. We infer that the Recon3 *MMN* contains the genes common to the metabolic networks of all human cells, tissues, or organs[28]. For all that, we found that when we used the *KEGG Orthology* information to compare the *MMN*s from all the species we examined, the *NED* genes showed a general, if inexact, taxonomic specificity (see Fig. 4).

We followed this up by performing a more precise analysis across species through a reciprocal BLASTp analysis to show the orthologs of the *NED* genes that can be found in entire reference genomes. Orthologs of proteins in other species were identified with best reciprocal Blast hit, using the Biopython pipeline developed by Nikolaidis et al. [29] for core proteome analysis. More specifically, the *S. cerevisiae* proteome was set as the reference proteome that is used to perform best reciprocal BLAST searches against all the other proteomes of the set. Figure 4 summarises these results, with the size of circles being proportional to the fraction of all the genes in the models having an ortholog in the genomes in the columns. The colour scale is related to the fraction of *NED* genes having an ortholog. In the vast majority of cases (558/589), the *NED* genes have more orthologs than the other genes and, in 385 cases, this difference is statistically significant (*p*-value < 0.05, Fisher's exact test). This plot makes clear the similarities across individual taxa. This is particularly true of the yeasts with their compact eukaryotic genomes.

A specific comparison between the very distantly related yeasts, *Sz. pombe*, and *S. cerevisiae* allows us to make an important general point. One of the *Sz. pombe* orthologs of the Magnificent Seven *NED* genes, found in all of our *S. cerevisiae MMN*s, is an essential gene and deletants of four more reduce the viability of stationary phase cells of

*Sz. pombe*[30]. Gene essentiality is context dependent, that context including both the growth environment and the genetic background. Liu et al. [31] conceive of essentiality as a quantitative trait that incorporates not only viability, but also evolvability. In this report, we show that *NED*ness is also a quantitative property that can be measured either as the proportion of *MMN*s that retain a given gene or in terms of a gene's impact on the global efficiency of the metabolic network.

We have introduced the new functional category of *NED* genes defined as the most retained non-essential genes in the *MMN*s. *NED* genes are more evolutionarily conserved across species than the other metabolic genes. In *S. cerevisiae*, these genes have other properties, such as an increased number of genetic interactions and a much lower number of paralogs; their gene products have more protein-protein interactions than even those of essential genes.

While our data appear self-consistent, it is important to verify our predictions by direct experimental tests. Published experimental data give some indication of the validity of our predictions. For instance, our pipeline correctly predicted some 80% of the core essential genes defined by van Leeuwen et al. [17]. Moreover, our success rate may well be even higher due to both the limited size of the van Leeuwen screen and some of the specific details of their experimental approach, including the use of temperature-sensitive alleles in intermediate steps of the procedure (J van Leeuwen, personal communication). In addition, according to an experimental survey of the growth kinetics of single-gene deletion mutants of *S. cerevisiae* (Warringer et al. [32]; Jonas Warringer, personal communication) single-gene deletions of *NED* genes show significantly reduced growth rates than non-*NED* genes (*p* = 4.38e-4; Mann-Whitney) but there was no significant difference in either the length of the lag phase or the biomass yield between the two classes of genes.

However, the impact of single-gene deletions is necessarily small and the minimisation of the metabolic network involves the deletion of many genes. Therefore, in order to validate our pipeline predictions, we constructed yeast strains carrying multiple deletions of either the Magnificent Seven genes or non-*NED* genes (using a random selection of candidates, see Methods) and measured their growth. Moreover, we

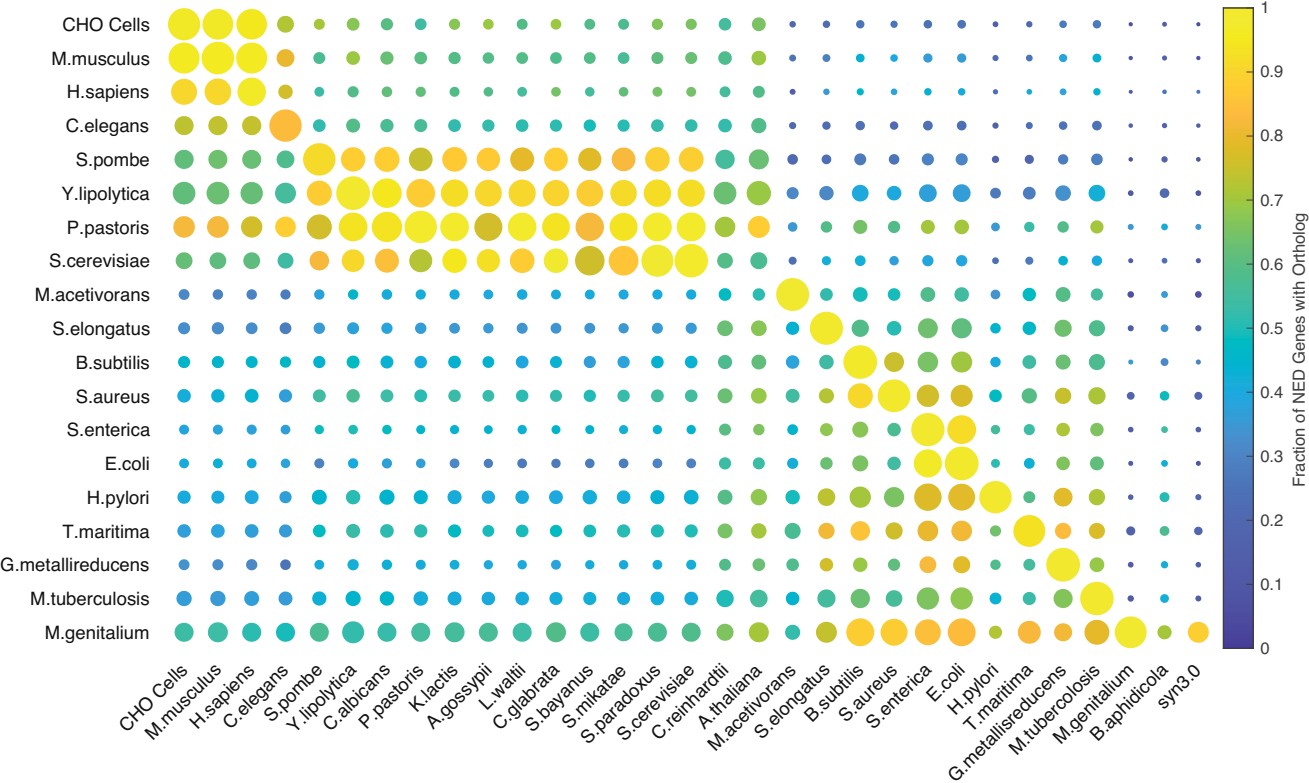

**Fig. 4 | Results of reciprocal BLASTp of the protein products of the genes in the models (rows) and some reference genomes (columns).** The size of circles is proportional to the fraction of genes in the model having an ortholog in the corresponding genome, while the colour scale is related to the fraction of *NED* genes having an ortholog. The *NED* genes have higher values in almost all the cases, with a majority of the differences being statistically significant (see text and SI; Mann-Whitney two-sided test used for statistical significance). Source data are provided as a Source Data file: Fig. 4 Source Data.xlsx.

## Table 1 | Multiple-deletion strains

| Class of genes | Double mutant | Triple mutant | Quadruple mutant |
|---|---|---|---|
| Magnificent Seven combinations | **ynk1Δ/gpt2Δ** | **ynk1Δ/gpt2Δ/tps2Δ** | |
| | **gpt2Δ/tps1Δ** | ynk1Δ/gpt2Δ/tps1Δ | |
| | **gpt2Δ/cho1Δ** | gpt2Δ/tps1Δ/tps2Δ | |
| | **ynk1Δ/cho1Δ** | gpt2Δ/cho1Δ/tps1Δ | |
| Magnificent Seven triple mutant (*ynk1Δ/gpt2Δ/tps2Δ*) + Δ *ARO* genes combinations | | | ynk1Δ/gpt2Δ/tps2Δ/aro1Δ |
| | | | ynk1Δ/gpt2Δ/tps2Δ/aro2Δ |
| | | | ynk1Δ/gpt2Δ/tps2Δ/aro7Δ |
| Non-*NED* combinations | **chs1Δ/dnf3Δ** | **chs1Δ/dnf3Δ/htd2Δ** | **chs1Δ/dnf3Δ/htd2Δ/coq5Δ** |
| | **chs1Δ/htd2Δ** | **chs1Δ/dnf3Δ/suc2Δ** | **chs1Δ/dnf3Δ/htd2Δ/zta1Δ** |
| | **dnf3Δ/suc2Δ** | **suc2Δ/chs1Δ/htd2Δ** | **chs1Δ/dnf3Δ/suc2Δ/coq5Δ** |
| | **dnf3Δ/htd2Δ** | | **chs1Δ/dnf3Δ/suc2Δ/zta1Δ** |
| | **suc2Δ/chs1Δ** | | |
| | **suc2Δ/htd2Δ** | | |
| Non-*NED* triple mutant (*chs1Δ/dnf3Δ/suc2Δ*) + aro7Δ combination | | | **chs1Δ/dnf3Δ/suc2Δ/aro7Δ** |

Viable constructs (able to grow on YPD-agar) are given in **bold**. Note that genes *aro1*, *aro2*, and *aro7* are *NED* genes that are invariably found in *MMN*s selected under aerobic conditions.

tested the fitness of both non-*NED* and Magnificent Seven mutants in combination with genes which display *NED* behaviour only in aerobic conditions (*ARO* genes). In total, we succeeded in constructing 19 of the 25 strains that we designed (Table 1). These included four double deletants and one triple deletant involving Magnificent Seven genes. While all the triple and quadruple deletion combinations involving non-*NED* genes were readily achieved, for the Magnificent Seven genes only one triple mutant could be obtained and no quadruple mutants; Table 1. Overall, 11 combinations involving the Magnificent Seven genes were attempted of which only 5 were viable. On the other hand, all 14 deletion combinations involving non-*NED* genes were achieved (Table 1).

We performed growth assays on solid and in liquid media (YPD and SD). On SD, two double mutants involving the Magnificent Seven genes did not grow and one was severely impaired (Supplementary Fig. 4C), while all the non-*NED* combinations of deletions had a more

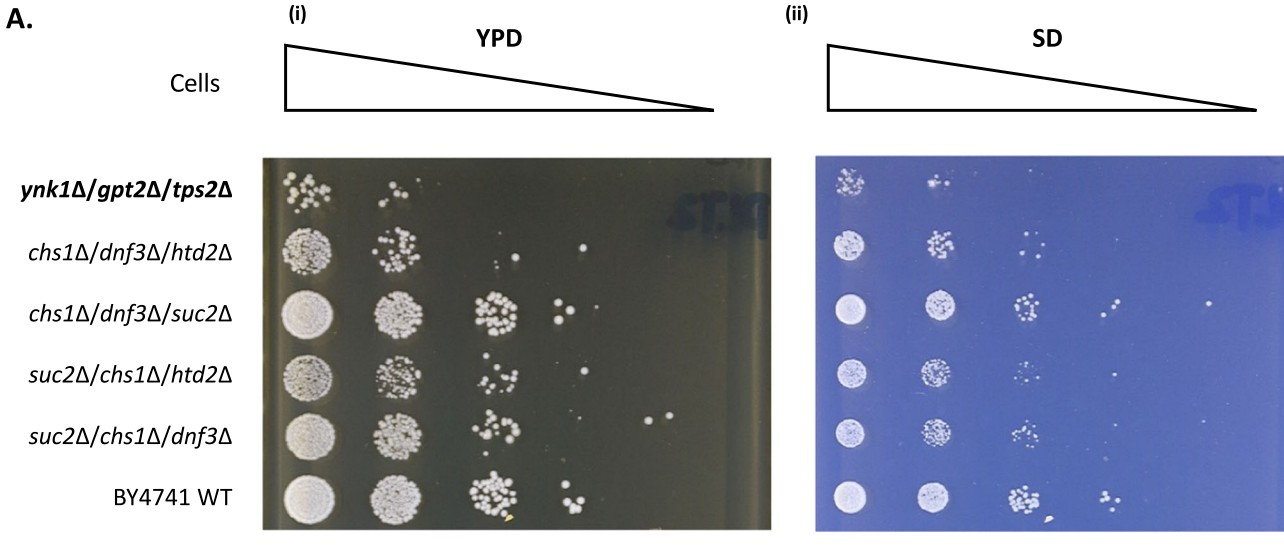

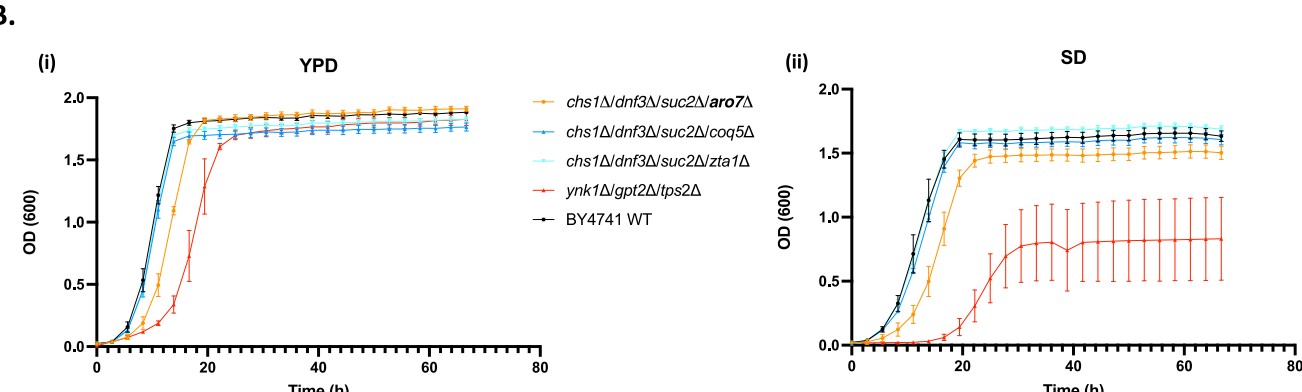

**Fig. 5 | Growth characteristics of multiply deletant strains. A** Spot test assay of: Magnificent Seven gene triple mutant (*ynk1Δ/gpt2Δ/tps2Δ*), non-*NED* gene triple mutants (*chs1Δ/dnf3Δ/htd2Δ, chs1Δ/dnf3Δ/suc2Δ, suc2Δ/chs1Δ/htd2Δ, suc2Δ/chs1Δ/dnf3Δ*), and BY4741 WT, on (i) YPD; and (ii) SD. **B** Growth curves of quadruple mutants and Magnificent Seven triple mutant. The *NED* gene *ARO7* (bold) and the non-*NED* genes *COQ5* and *ZTA1* were deleted from the same triple mutant background (*chs1Δ/dnf3Δ/suc2Δ*). Strains were grown in YPD rich medium (i) and SD (ii).

See Supplementary Figs. 4–6 and Supplementary Data 7 for a complete characterisation of growth of WT and all multiply deletant strains in liquid culture. Data are presented as mean values +/- SD (standard deviation), with 3 technical replicates and 2 biological replicates (except for *chs1Δ/dnf3Δ/htd2Δ/zta1Δ*, for which only one transformant was recovered). Source data are provided as a Source Data file: Fig. 5 and SI Fig. 4–6 Source data for experimental parts.xlsx.

modest fitness deficiency (Supplementary Fig. 4C). Both solid and liquid growth assays confirmed that the triple mutant involving the Magnificent Seven genes had a much greater growth defect compared to the non-*NED* triple mutants (Fig. 5A and Supplementary Fig. 5) or non-*NED* quadruple mutants (Fig. 5B and Supplementary Fig. 6). Moreover, the quadruple mutant that combined deletions in three non-*NED* genes and *ARO7*, a *NED* gene that is invariably found in *MMN*s selected in aerobiosis, was found to have a greater growth defect than that any of the quadruple mutants that involved only non-*NED* genes, although not as severe as that of the Magnificent Seven triple mutant (Fig. 5B).

Measured growth characterstics of WT and all multiply deletant strains are given in Supplementary Figs. 4–6.

It should be noted that, for each successive deletion event, we had a positive selection for the gene replacement involved (see Methods). Thus, the fact that we were only able to construct one triple mutant and no quadruple mutants among the the Magnificent Seven genes indicates that these gene-deletion combinations are either lethal or so sick that they cannot recover after transformation. Similarly, the construction of quadruple mutants involving three Magnificent Seven

Genes plus either *aro1Δ* or *aro2Δ* or *aro7Δ* were inviable, whilst a quadruple mutant involving deletions in three non-*NED* genes and *aro7Δ* was viable and easily constructed. The successful isolation of all mutants attempted among the non-*NED* genes demonstrates that our operational ability to perform gene deletion is robust. So, this observation combined with the fitness data for the viable mutants of Magnificent Seven genes (Supplementary Fig. 4) suggest that mutant combinations of the Magnificent Seven genes are either lethal or very sick or unfit, precisely because there are strong negative genetic interactions between them.

Taken together, such experimental data explain why the Magnificent Seven genes are retained in all the *MMN*s. In other words, our modelling and simulation studies have revealed these higher-order interactions and they have been confirmed by in vivo experiments, thus providing a substantial validation of our pipeline approach to the design of *MMN*s.

The results we have presented have important implications for synthetic biologists trying to design *MMN*s (or, indeed, synthetic organisms) either by the deconstruction of extant metabolic networks by gene deletion or by the de novo design of an entirely artificial

network[3,26]. The *NED* genes play a major role in determining the global efficiency of both the metabolic network and of the genetic interaction network of metabolism genes. This finding reinforces our contention that genes which are Network Efficiency Determinants (*NED*) are an important new functional class that need to be considered when synthetic biologists design new strains and organisms, they may also provide important signposts for those studying the evolutionary origins of metabolism.

## Methods
### Model
The base of our study is the genome-scale metabolic model of *S. cerevisiae*, as reported in Supplementary Table 1. The genes annotated as essential were not considered in our minimisation procedure.

### FBA, FVA, pFBA
The genome-scale models are evaluated in all the steps using the well-known approach of *Flux Balance Analysis*[5]. The core of the method is the stoichiometric matrix $S$, where every row represents a metabolite, and every column represents a chemical reaction. The elements of the matrix are the stoichiometric coefficients of the chemical reactions, and the matrix is *sparse* since most of the elements are null. The matrix $S$ defines a set of constraints for the model in the compact form $Sv = 0$, which are a representation of the mass balance equations in steady state, where $v$ is the vector of all the fluxes through the reactions of the model. These constraints define a solution space of an optimisation problem with the maximisation of the flux $v_{bio}$ through a pseudo-reaction of the model simulating the growth rate of the cell. (In the FBA models there is one biomass pseudo-reaction, also known as biomass objective function[33].) We also implemented a simple approach similar to the *Flux Variability Analysis*[5] to define the minimal bounds in the two minimal media used for the model of *S. cerevisiae* (see Media Definition sections). FVA is a very accurate way to determine the maximum and minimum values of all the fluxes that will satisfy the constraints and allow for the same optimal objective value. Basically, FVA is a method to determine the *range* of possible reaction fluxes that still satisfy, within some optimality factor, the original FBA problem.

For the reaction flux values that we used for the analysis using complex network theory, we implemented the *parsimonious Flux Balance Analysis*[5] approach, that returns a fluxes distribution minimising the sum of squares of all the fluxes. The simple assumption behind this method is that a cell, under exponential growth, tends to adopt a behaviour that requires the lowest flux through the metabolic network. The results of this approach are ideal when considering a global behaviour of the network since it avoids unrealistic values of some of the fluxes, e.g. when they are involved in futile cycles within the network. We do not further describe these methods here, referring the reader to the cited literature for all the details.

### KnockOut simulation
In addition to the reactions, the genome-scale metabolic model also includes information on the genes of the organism. An array with all the protein-coding genes is present in the structure, and many of the reactions are related to a logical rule involving those genes. The logical connectors AND OR simulate the isoenzymes and protein complexes. The presence or absence of the genes in the model is represented by a logical array, with a True value corresponding to a present gene and vice versa. Every different solution considered in our analysis can therefore be represented by a logical array. Hence, the wild type is the all-True array, and every single mutation is the switch of one of the logical values of the array. Using the logical values in the array, the rules of the reactions are evaluated. Some of the rules can have a false value as a result of the simulated KOs and the corresponding reactions will then be excluded from the model when the network is evaluated with the FBA.

### Media definition
The exchange reactions of a model simulate the external environment of the cell and have a big impact on the behaviour of the metabolic model, both in terms of the numerical value of the rate of biomass formation (growth rate) and the predicted fluxes of the chemical reactions.

For *S. cerevisiae*, we used 6 different simulated media conditions (see Supplementary Table 7 and Supplementary Data 3), 3 for aerobiosis, SD, Minimal and YPD media, and 3 for anaerobiosis, being the version of the former with the addition of fatty acids[34]. To simulate the growth in the absence of oxygen we also removed the haem requirement in the artificial growth rate[35]. SD and YPD definitions are based on literature; we used the *SD* medium setting defined in Labhsetwar et al.[36]. and set the glucose uptake rate bound to 15 *mmol/gDW/h*; we used the YPD medium definition from Harrison et al.[37].

Alongside the media taken from literature, we also used a simple procedure (see Algorithm 1 in Supplementary Information), inspired by the one used for the by Pal et al.[15], that could automatically define a new minimal medium, given a minimum value of the biomass function to be guaranteed. The idea is to set all the exchange reaction bounds as unconstrained ( $= 10^3$ ) at the beginning, and then sequentially set one of them, chosen at random, to zero i.e., removing that compound from the simulated medium. The change is kept if the predicted biomass is well above the minimum, otherwise it is restored, and the procedure is repeated. If, after a number (*tol*) of attempts, an exchange reaction to be removed is not selected, a function implementing an exhaustive procedure is called. All the residual removals are tested in it and, if a feasible one is found, is returned to the main procedure, otherwise the procedure ends and returns an empty array, since no further exchange reactions bounds can be set to zero. The glucose, oxygen (if aerobic), and water exchange reactions bounds are fixed and not considered. Finally, we have defined a minimal medium composition entirely generated by this simple algorithm developed for the task, and the corresponding anaerobic version. After this, a new procedure is called in order to narrow the possible uptake fluxes, using a simple model-oriented approach to redefine the bounds of the exchange reactions. Using an approach analogous to FVA, we determined the minimum flux for each exchange reaction that ensures the Growth Rate value obtained by FBA as the new bound of that exchange reaction, i.e. limiting the amount of nutrients provided to the metabolic network. All the media use glucose as the principal carbon source; water and oxygen (when present) uptake bounds are left unconstrained in all the media.

### Algorithm
In this section, we describe the evolutionary algorithm used for obtaining the minimal metabolic network. The procedure is the same for all the media. In Algorithm 2 in Supplementary Information the main procedure is described. The idea is an evolutionary algorithm which iteratively improves the initial population for maximising the number of KOs. Every point of a population represents a candidate solution, i.e. a set of genes that are knocked out. They are represented by a logical array (as defined in KnockOut simulation), with every value corresponding to a gene in the model. If the value is equal to 0 (false), the gene is still present in the model, otherwise it is knocked out.

The initial population points are all wild-type strains, i.e. all-zeros arrays. During the procedure, the genetic operator function (see Algorithm 3 in Supplementary Information) is the first function to be called. It selects a new possible knockout over all the remaining active genes in the strain and it evaluates the new Growth Rate. If the Growth Rate satisfies the constraint the change is kept, and the new point will enter in the new population, otherwise the searching is repeated till a feasible knockout is reached or a maximum number of trials (10) are performed. The constraint over the Growth Rate is such that the new

strains cannot have a predicted value smaller than 99% than that of the Wild Type grown under the same conditions.

The genetic operator constructs an offspring for every parent point in the population and all these new points constitute the offspring population. The union of this and the parent population is then sorted using the *sortPop* function (Algorithm 4 in SI). The idea of the sorting is that the points that were improved with a new knockout should be discarded. In order to do this the Hamming Distance of all the couples of points is evaluated. The Hamming distance between two solutions $p$ and $q$ in our case can be defined as the lowest number of changes (from 1 to 0, or vice versa, in the logical array of the solution) that are necessary to obtain the solutions $q$ starting from the solution $p$. If two points $p, q$ P have a Hamming Distance equal to 1, and $p$ has a higher number of KO than $q$ has, i.e. $nKO(p) = nKO(q) + 1$), we say that $p$ *dominates* $q$. This definition lets us consider the non-dominated points as the ones to be kept. We assign them a number 1, corresponding to the first *front;* the procedure is then repeated ignoring the points already labelled, finding the points of the second front and so on. The points with the lower front value are then preferred over the others, but we also need a criterion of selection among the same front. This criterion is based again on the Hamming Distance. For every point we assign a value, i.e. the mean of the Hamming distances between it and its closest 10 neighbours (always in terms of the Hamming distance). In the same front we select the points that have the greater value so defined; this helps maintain a high diversity degree in the population.

All the points also have a feature while in the population, their ages. The age can be defined as the numbers of the last generation in which the point has been present. If a point was not improved during the last generations of the algorithm, it is a candidate to be a minimal solution. If this is the case, there are no more knockouts that can be selected for the strain that satisfies the Growth Rate constraint; there is no point then in keeping this point in the population. The age is then used as the variable of a sigmoid function representing the probability of a solution to be discarded from the population (see Algorithm 5 in Supplementary Information). If the point has to be replaced, the *Backtracking* function is called. This function simply selects a random set of genes knocked out in the strain and turns them on, going up in the search tree (not necessarily to a node that has already been visited). If the new strain satisfies the Growth Rate constraint, then it will be the replacement in the population.

At the end of the algorithm a post-processing procedure is launched to identify the minimal solutions found by the algorithm. Following the definition of non-domination, given for the sorting of the population, the solutions that are not dominated are selected. All of them are then tested using an exhaustive procedure, similar to the one used in the definition of the new media. All the single KO of the genes still active in the string are tested, to ensure that there are not any other genes that can be turned off while still satisfying the biomass constraint. If this is the case, the solutions can then be defined as *minimal* and be included in the results returned by the procedure.

## Parameters and complexity

The usual parameters we used in our experiments are a population size equal to 100 and a maximum number of generations equal to 5000. The theoretical maximum number of candidate solutions explored during the procedure is then equal to 500,000 (potentially to be multiplied by 10 if we consider the maximum number of trials in the genetic operator). The actual number of points explored is lower than this, as part of these points are found during the procedure and are far from a possible minimal network. In the *S. cerevisiae* experiments using the *yeast 8.3.1* model[8] (which includes 3949 reactions, 2680 metabolites and 1133 genes; 157/1133 genes are essential) the mean of the unique solutions explored is close to 350,000. Among all these, the number of unique Minimal Networks found by the evolutionary

algorithm is 750-800. An important remark is that the dimension of the raw solutions space, in the case of the *yeast 8.3.1* model, is theoretically equal to $2^{976} \approx 10^{293}$; this is incredibly complex and infeasible for an exhaustive search. For the choice of algorithm parameters (e.g., number of iterations and number of metabolic networks at each iteration), a compromise was adopted that allowed a reasonable number of metabolic networks to be evaluated by reducing the convergence process of the algorithm.

## Frequency and redundancy analysis

Once the algorithm returns the MNs found by the algorithm, a set of analyses was performed. First, we considered the frequency of knockouts for every gene. Three sets of genes are defined: *i)* genes that are always knocked out in every *MMN*; *ii)* genes that are knocked out in some of the *MMN*s but are present in others; *iii)* genes that are always active in the solutions. (Another category would be the essential genes, which are not considered by the algorithm.) We can also rename this knockouts frequency analysis as a measurement of the redundancy of some of the genes; in particular, we can consider as more or less redundant the genes that have a higher or lower knockouts frequency in the *MMN*s, respectively.

The analysis was also extended, for *S. cerevisiae*, to include all the *MMN*s in aerobic or anaerobic conditions altogether, or in both of them (see Supplementary Tables 1–6). In general, the genes that are (almost) always present in the solutions are the most interesting because of the important predicted role they have in the metabolic networks. Moreover, if they are not marked as essential (as in the yeast case), that could shed new light on the importance of the gene in metabolism and/or in the cell, or either a false positive due to the approximations that are included in the model.

## Complex network analysis

The metabolic network given by the model and the set of genes present in it, can be visualised as a graph and can be used as a complex network. There are several ways to perform this task, such as a hypergraph or simple directed graph. We considered the representation of the network as a bipartite directed graph. Every metabolite and every reaction of the model becomes a node in the graph, while the edges follow the stoichiometry of the reactions, e.g. if the metabolites $m1$, $m2$ are a reagent and a product of the reaction $r$ respectively, two directed edges $(m1, r)$, $(r, m2)$ are added to the graph. The metabolites nodes are then not directly connected by an edge in the graph, and the same for the reaction nodes, hence the bipartite graph. Using the predicted reactions fluxes, we can add the weights on the edges of the graph, taking care of multiplying the flux by the metabolite stoichiometric coefficient in the reaction. If we consider the weights, some of the reactions will not be active (i.e. they have a flux equal to 0) and so the corresponding nodes and edges are not considered in the network, which is then simplified. Once we have defined the weighted graph in this way, we can use some basic measures and quantities defined in complex network theory to analyse the behaviour of the network[38]. Specifically, we considered the cumulative distributions of edge weights and node degrees for a description of the network.

The other networks considered in our study are the genetic interaction networks involving the genes present in the genome-scale model, with the edges between genes representing a positive or negative interaction. In this case we evaluated the nodes' importance using basic measures from complex network theory, such as the Latora-Marchiori efficiency[20], the degree of the nodes or the measure of betweenness. The efficiency is a simple measure based on the paths between nodes; given a graph $\boldsymbol{G}$, the efficiency is defined as:

$$E(\boldsymbol{G}) = \frac{\sum_{i \neq j \in \boldsymbol{G}} \epsilon_{ij}}{N(N-1)} = \frac{1}{N(N-1)} \sum_{i \neq j \in \boldsymbol{G}} \frac{1}{d_{ij}}$$

Where $N$ is the number of nodes of the graph, $i,j$ are two nodes and $d_{ij}$ is the length of the shortest path between them. Born as a description of how efficiently the information is exchanged in a network, the efficiency is linked to the diameter of the graph and is easy to compute even for large graphs, but despite its simplicity it is a powerful instrument to evaluate a network. Instead, the degree of a node is simply defined as the number of edges that are incident to it, while the betweenness is defined considering the set of all the shortest paths for all the possible pairs of nodes in the network; the betweenness of a node is then defined as the number of these paths that pass through the node. Compared to the degree, the betweenness is able to capture the "bottlenecks" nodes, connecting (for example) two different sections of the graph with an obligatory route, but not necessarily having a large number of connections. On the other hand, the marginal nodes in terms of shortest paths across the network, but with a significant number of connections, are more likely to be highlighted by the degree metric. For each measure we considered an average analysis and for degree the cumulative distributions and the frequency of nodes degree, with the power laws obtained using a maximum likelihood estimator on the $\gamma$ values.

## Code implementation and machine settings

The main algorithm and the analysis codes[39] were developed on MATLAB software by MathWorks, Inc. The main procedure was run on *MATLAB r2017a* executed on a HPC cluster node with 2 Intel Xeon Skylake 6142 processors, 2.6 GHz 32-core and 192GB RAM, O/S Scientific Linux 7. The analyses were mainly performed on *MATLAB r2018b* executed on a laptop with an Intel i7-6770HQ processor, 2.6 GHz 4-core and 16GB RAM, O/S Windows 10 by Microsoft. The Linear and Quadratic Programming Problems were solved using the MATLAB interface of the Gurobi software, v. 9.0.0.

## In vivo validation experiments
### Random selection of non-*NED* genes

Non-*NED* genes were selected with the random initialisation function of Python3 (using random initial seeds) from the list of non-*NED* genes.

**Construction of double, triple and quadruple mutants.** Multi-gene deletants were constructed by deleting genes of interest sequentially from single-gene deletion mutants (carrying *kan*MX) via multiple rounds of target gene replacement using either antifungal resistance cassettes (*nat*NT2, *hph*MX6) or a *URA3* marker[40]. Successful transformants were confirmed via colony PCR. All primers used for construction of the deletion cassettes and confirmatory PCRs are listed in Supplementary Data 6.

**Growth assays.** For liquid growth assays, overnight cells were inoculated into YPD or SD in 96-well microplate at a starting $OD_{600} = 0.1$. Growth was measured using a microplate reader (FLUOstar Omega, BMG Labtech) including three technical replicates for each mutant. For the spot assays, overnight cultures were spun down and diluted with water to reach a final OD600 = 4. Ten-fold serial cell suspension with concentrations ranging from $OD_{600} = 0.4$ to $OD_{600} = 4 \times 10^{-6}$ were spotted on the agar plates (YPD and SD) and pictures of growth were taken after 48 h. Uncropped pictures are reported in Supplementary Fig. 7.

## Data availability

Source data are provided with this paper. Code available on GitHub: https://github.com/GiuseppeNicosia1/MinimalNetwork_CompleteCode. https://zenodo.org/records/13362420. JSON Files for metabolic map construction: https://github.com/GiuseppeNicosia1/MinimalNetwork_CompleteCode/blob/main/escher-metabolic-maps.zip Source data are provided with this paper.

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

## Acknowledgements

We thank Charles Boone (Toronto), Balazs Papp (Szeged), Jolanda van Leeuwen (Lausanne), Soukaina Timouma (Manchester), Jonas Warringer (Gothenbug), and Valerie Wood (Cambridge) for helpful discussions. D.D. is supported by the Future Biomanufacturing Research Hub (Future BRH), funded by the Engineering and Physical Sciences Research Council (EPSRC) and Biotechnology and Biological Sciences Research Council (BBSRC) as part of UK Research and Innovation (grant EP/S01778X/1). G.J. was supported by a Biotechnology & Biological Sciences Research Council (UK) grant no. BB/N02348X/1 to SGO as part of the IBiotech Program, and by the Industrial Biotechnology Catalyst (Innovate UK, BBSRC, EPSRC) to support the translation, development and commercialization of innovative industrial biotechnology processes. G.N. and G.J. thank the University of Catania for leave to pay extended research visits to Cambridge.

## Author contributions

Conceptualisation and supervision: G.N., S.G.O. and D.D. Algorithm development: G.J. and G.N. Pipeline and Monte Carlo experiments: G.J. Complex network analysis V.L., G.J. and G.N. Bioinformatic analyses: G.D.A. and S.G.O. Design of multiple mutants: D.D. and S.G.O. Construction and testing of multiple mutants: T.Q. and D.D. Statistical analyses and data visualisation: G.J., G.D.A. and T.Q. Interpretation of results: G.J., T.Q., G.D.A., D.D., S.G.O. and G.N. Writing of manuscript: G.J., D.D., G.N. and S.G.O. Revision of manuscript: G.J., G.D.A., V.L., D.D., S.G.O. and G.N.

## Competing interests

The authors declare no competing interests
