## [Peer Review File · Nature Communications]

Minimisation of metabolic networks defines a new functional class of genesREVIEWER COMMENTS

Reviewer #1 (Remarks to the Author):

Will the work be of significance to the field and related fields? How does it compare to the established literature? Such MMNs along with the mandatory genes could be useful in obtaining more readily an overview of yeast behavior by focusing the attention of the essential genes. Typically the yeast models (several published in Nature Communications) become more complex as one tries to include more reactions its approximately 6000 genes can perform, so eliminating redundancies and alternate pathways could be useful in synthetic designs.

Does the work support the conclusions and claims, or is additional evidence needed? Are there any flaws in the data analysis, interpretation and conclusions? Do these prohibit publication or require revision? Not at the moment

Is the methodology sound? Does the work meet the expected standards in your field? Is there enough detail provided in the methods for the work to be reproduced? Not at the moment.

The paper would be vastly improved by the considering the following comments;

1. I may have missed it but what is the doubling time for the wild type under all the various medium compositions? It seems you use a rule that any gene knockout producing a drop of 1-10% in the growth rate of the wild type to identify the minimal metabolic networks, but how the mandatory genes are chosen is not clear. Under various medium conditions, I have seen the term quasi-essential used in the context of genes that are not essential, but are needed to maintain a given doubling time within some reasonable range. Is this the same term as what you are calling "mandatory"?

2. Many interesting numbers used in your analyses seem to be missing: What are the number of genes in the yeast 8.3.1 model? While essentiality can be context and media dependent, but what are the number of essential genes you are assuming in your algorithms? And give the reference.

3. Provide a Discussion Section: Representation of the data in the Figures and Tables in main manuscript and extended data files could be much improved and results summarized in a Discussion section. For example,

Tables 2 – 5: contain a lot of information and comparisons, but the information would be more useful if also presented as metabolic map figures and summarized in a Discussion section which is missing.

The metabolic maps (drawn with software like Escher or CellDesigner) showing the largest number of mandatory genes say for the SD medium or largest number from combined anaerobic and aerobic results would allow the reader to immediately understand and assess the results. The authors wrote that there are between 750-800 minimized metabolic networks (MMN), so it should be possible to make metabolic maps for each compartment using the genes from all aerobic and anaerobic conditions. Even better would be to compare sections of the yeast 8.3.1 maps with mandatory genes in bold.

4. Give the MMN as JSON or SBML files which would allow the use of Escher or some of software to present the metabolic network in Yeast 8.31 and then gray out the links that can be eliminated and still stay within the 1-10% grow rate cut-off.

5. Table 6 – What does ribosome compartment mean? 9 copies of the rRNA?

Minor comments:

Line 54: I'd be interested to see if the algorithm agrees with the literature on which genes are essential. If you remove the constraint that they can't be removed, are they kept anyway?

Line 65: Please provide references for the medium compositions here.

Line 69: I might have missed this, but is it known experimentally that yeast grows at the same rate in all conditions you tested? Did you optimize for the same doubling time in all conditions?

Line 155: Wrong subfigure references, or subfigure labels are incorrect in the figure.

Check the references. A few are missing the year.

Reviewer #2 (Remarks to the Author):

The manuscript submitted by Jansen and colleagues reports the development of a computational pipeline to determine minimal metabolic networks (MMNs) with predicted growth outcomes above a certain threshold relative to the wild type. The pipeline starts with a metabolic model and a list of genes that are considered essential in the literature and cannot be deleted. An evolutionary algorithm is then used to iteratively delete genes and eventually reach the MMN. By working with the *S. cerevisiae* metabolic network in six different growth conditions, the authors have noticed a category of "mandatory" genes that are not essential but appear difficult to exclude from the MMN under most if or all conditions. More specifically, seven genes (named the magnificent seven by the authors) are always present in the MMN. The authors indicate that mandatory genes have more genetic and protein-protein interactions and a greater impact on the predicted growth than other genes taken at random in *S. cerevisiae*.

Overall, I found the topic of this manuscript to be interesting, and I believe that future work in synthetic biology will be aimed at building MMN. However, I have questions and concerns about the work that I think would be important to address:

I was disappointed that the predictions were only performed for one organism, *S. cerevisiae*. I appreciate the fact that six different conditions were evaluated but this limits the conclusions that can be drawn about mandatory genes. The authors mentioned that MMN for other organisms were also analyzed but that the results would be described elsewhere. This would have been a nice addition to understanding the general importance of mandatory genes, their conservation across species, and the type of reaction that they catalyze.

I doubt that "mandatory" is the best term for this group of genes that are rarely omitted from MMN. Mandatory can bring confusion with essential, whereas in this case mandatory genes can be deleted individually (experimental data is demonstrating this) but not when metabolic reduction is brought to an extreme.

It is unclear after reading the manuscript how their conclusion is affected by the biomass objective function used by the model. The authors acknowledge this limitation and state that the cell biomass composition depends on old and potentially unreliable data. I wonder what the impact of modifying the biomass objective function would be, for example by using a computational approach that refines the exact components that the cell should make (PMID: 31009451).

Given that the list of mandatory genes could vary between species and potentially between conditions (for example if the biomass function varies), I am not convinced that the seven mandatory genes deserve to be called "magnificent". This leads the readers to think that these genes are special while they are not more important than strictly essential genes for cell growth. Finally, while I understand the difficulty of generating a cell with an MMN and that the authors refer to published experimental data (e.g. genetic or protein-protein interactions), having a few experiments trying to validate the difficulty of knocking-out combinations of mandatory genes (especially a few of the magnificent seven) would provide an important support for their conclusions. Otherwise, the conclusions remain purely based on computational predictions for which the validity remains hard to accurately estimate.

On a more minor note, I would like to tell the authors that Figure 1 can be a bit confusing since the pale blue that overlaps with the purple is very similar to the slightly darker blue used in the same panel. In addition, the Extended Data Table 1 is difficult to read because of the formatting of the last column on the right side of the page.

Reviewer #3 (Remarks to the Author):

Jansen et al. reported an *in silico* approach (based on an evolutionary algorithm) to reduce the number of genes in a given metabolic model without compromising biomass production. They applied it to a *Saccharomyces cerevisiae* metabolic model. The authors referred to the resulting metabolic network models as minimized metabolic networks (MMNs). The authors applied their approach to simulate effect of gene deletions in three different growth media in aerobic and anaerobic conditions. Remarkably, by analyzing the frequency of each metabolic gene in the resulting MMNs, authors proposed a new class of genes, mandatory genes, that is different from the essential gene category and seemed to be central for the proper flow of information in the resulting MMNs. Therefore, this work could contribute to our understanding of the general principles of metabolic networks while allowing us to re-evaluate the importance of potentially overlooked metabolic genes in *S. cerevisiae*. Although an appealing idea, there are several concerns that should be addressed by the authors in order to offer support to their claims (see comments below). In addition, the authors could gain additional insights by taking a closer look at the resulting MMNs.

Major comments

1. One of the main concerns with the study is the lack of model performance assessment and experimental validation. It may not be possible to validate predicted biomass of the MMNs (with dozens of deleted genes), however authors should evaluate the accuracy of their metabolic network models to predict single gene deletions in the different growth contexts. They could achieve this by:
 - Comparing the set of essential genes predicted by their metabolic models for each growth condition with available experimental data.
 - Comparing biomass predictions for single gene KOs with data of growth effect of gene deletions.
 - Comparing experimental growth of WT *S. cerevisiae* in the six growth conditions with the predicted biomass for WT.
2. Similarly, the proposed class of mandatory genes are based exclusively on computational predictions, no experimental support is given (not even for the magnificent seven genes). So, authors need to offer experimental support for this new gene category. To do so, authors could experimentally evaluate and compare the growth effect of single gene deletions of multiple mandatory and non-mandatory genes while comparing the biomass production with their model predictions.
3. The authors need to revise their list of essential genes in a context-specific manner, and re-run their analyses with the curated list of essential genes. During a quick literature review of the magnificent seven genes, I found that several of them (e.g., TPS1, TPS2, CHO1, ADE3) have been classified as essential in minimal growth medium (see for example Heavner & Price 2015- cited by the authors, and other references such as PMID: 26426067, 19525417). Based on those previous reports, it is not surprising that deletions of most of those seven genes were predicted to completely inhibit growth.

As the authors stated in L58-l60, the definition of essentiality depends on the composition of the growth condition, therefore authors need to carefully review their list of essential genes taking into account the conditions used in their simulations. Once authors have re-run their analyses, they can check if the new gene category they proposed is still present (see my comment # 2).
4. Given the immense space of possible combinations in the metabolic network (L478: 2^{976}), how reproducible are the results? If the authors run their simulations again, would they arrive to the same conclusions?
5. Additional questions that the authors could address to get additional insights:
 - Can the authors apply a different objective function (for example, ethanol production) and compare how different are the properties of the resulting MMNs and the corresponding mandatory genes?
 - How different are the fluxes of the reactions in the reduced networks with respect to the whole

WT metabolic network?

- Are there any emerging topological properties of the MMNs that distinguish them from other metabolic or biological networks?

Minor comments

6. Extended data table1

-What is the exact definition of MNs here? For example in row # 1, there are 787 MNs. Do you mean MMNs?

-In the same row #1, why do you use the nomenclature 81+4 mandatory genes instead of 85 genes?

7. L63-65: what was the point of generating an ad hoc minimal medium instead of using a standard minimal medium that would allow easier comparison with public data? Authors should consider replacing this condition with a standard minimal medium

8. L73: how many generations exactly? Please be specific.

9. Figure 1

- Panel A: are the shown histograms for the 10% or 1% biomass threshold?

- Panel B: It is unclear to me what the aerobic vs anaerobic represents. Specifically, how is different from the relevant comparison (e.g., top panel: SD aerobic and SD anaerobic)? Why is the yellow histogram different in each subpanel? What is each condition compared to when defining the fraction of shared reactions?

10. L102-103: do the authors mean higher number of transport-related genes in the SD aerobic model? It may help if authors highlight with a star or other symbol the transport-related categories in Extended Data Fig 3.

11. Extended Data Figure 2: It is unclear to me how the lyase activity category is related to the transport function. Can the authors explain this a little in the text?

12. L105-106: this sentence disagrees with Extended Data Figure 2a which shows a higher number of KOs in the SD anaerobic model with respect to the SD aerobic model.

13. L121-122: It seems that the authors defined the mandatory gene set independently for aerobic and anaerobic conditions. If that's the case you need to mention it. Or do you define the mandatory genes for each condition?

14. L141: the concept of biomass pseudo-reaction could be explained in more detailed. The FVA abbreviation was introduced here without any background. It is also unclear why authors were interested in the pseudo-reaction and not the biomass itself.

15. Algorithms #2 and #5 and the functions that authors referred to in the methods section are briefly described in the text but there is no additional information in the supplement.

16. A supplement figure describing the algorithm used to derive the MMNs may be useful for the reader.

REVIEWER COMMENTS – Jansen *et al.* NCOMMS-23-20540

Reviewer #1 (Remarks to the Author):

Will the work be of significance to the field and related fields? How does it compare to the established literature?

Such MMNs along with the mandatory genes could be useful in obtaining more readily an overview of yeast behavior by focusing the attention of the essential genes. Typically the yeast models (several published in Nature Communications) become more complex as one tries to include more reactions its approximately 6000 genes can perform, so eliminating redundancies and alternate pathways could be useful in synthetic designs.

Does the work support the conclusions and claims, or is additional evidence needed? Are there any flaws in the data analysis, interpretation and conclusions? Do these prohibit publication or require revision? Not at the moment

Is the methodology sound? Does the work meet the expected standards in your field? Is there enough detail provided in the methods for the work to be reproduced? Not at the moment.

The paper would be vastly improved by the considering the following comments;

1. I may have missed it but what is the doubling time for the wild type under all the various medium compositions? It seems you use a rule that any gene knockout producing a drop of 1-10% in the growth rate of the wild type to identify the minimal metabolic networks, but how the mandatory genes are chosen is not clear. Under various medium conditions, I have seen the term quasi-essential used in the context of genes that are not essential, but are needed to maintain a given doubling time within some reasonable range. Is this the same term as what you are calling "mandatory"?

For any given medium, the designed optimization algorithm identifies the set of mandatory genes. The algorithm is described in the Methods section while the related pseudo-code is presented in the Supplementary Methods section (referred to as Algorithm 2).

We define **mandatory** the genes that have the following 8 characteristics:

- 1) The mandatory genes are **not essential**.
- 2) The mandatory genes ensures both **viability and high growth rates**.
- 3) The mandatory genes are very **rarely eliminated in constructing a minimal metabolic network (MMN)**, suggesting that it is difficult for metabolism to be re-routed to obviate the need for such genes.
- 4) The removal of *mandatory* genes from the minimised metabolic network significantly **reduces its global efficiency**.
- 5) Bioinformatic analyses of the *mandatory* genes have revealed that not only do these genes have **more genetic interactions** than the bulk of metabolic genes
- 6) but their protein products also show **more protein-protein interactions**.
- 7) In yeast, *mandatory* genes are **predominantly single-copy** and
- 8) are **highly conserved across evolutionarily distant organisms**.

All these features confirm the importance of the *mandatory* genes to the metabolic network, including why they are so rarely excluded during minimisation.

The mandatory genes were validated with the new *in vivo* experiments included in the revised manuscript.

2. Many interesting numbers used in your analyses seem to be missing: What are the number of genes in the yeast 8.3.1 model? While essentiality can be context and media dependent, but what are the number of essential genes you are assuming in your algorithms? And give the reference.

The number of genes and the number of essential genes is both reported in the Extended Data Table 1 of the Supplementary Information (p. 1). We also added this information in the Methods section **Parameters and complexity** [see p 17, lines 577-590]

3. Provide a Discussion Section: Representation of the data in the Figures and Tables in main manuscript and extended data files could be much improved and results summarized in a Discussion section. For example,

Tables 2 – 5: contain a lot of information and comparisons, but the information would be more useful if also presented as metabolic map figures and summarized in a Discussion section which is missing.

The metabolic maps (drawn with software like Escher or CellDesigner) showing the largest number of mandatory genes say for the SD medium or largest number from combined anaerobic and aerobic results would allow the reader to immediately understand and assess the results. The authors wrote that there are between 750-800 minimized metabolic networks (MMN), so it should be possible to make metabolic maps for each compartment using the genes from all aerobic and anaerobic conditions. Even better would be to compare sections of the yeast 8.3.1 maps with mandatory genes in bold.

This is a lot of work, does not seem to add very much, and could only be presented in Supplementary Material. Indeed, such presentations may even be counterproductive since we have *quantified* the mandatory in terms of their impact on network efficiency. This cannot be conveyed by simply bolding the mandatory genes.

4. Give the MMN as JSON or SBML files which would allow the use of Escher or some of software to present the metabolic network in Yeast 8.3.1 and then gray out the links that can be eliminated and still stay within the 1-10% grow rate cut-off.

This request appears to be more appropriate than the previous one but involves the creation of six such maps, to cover all 6 conditions for which we have selected *MMNs*. Moreover, as we have been at pains to point out, there is no unique *MMN* for each condition and the definition of mandatory genes by defining the proportion of *MMNs* that contain the gene again *quantifies* their contribution to the network.

5. Table 6 – What does ribosome compartment mean? 9 copies of the rRNA?

The compartment designations derive directly from the yeast metabolic model 8.3.1 [ref. 13]

<https://www.nature.com/articles/s41467-019-11581-3>

This paper explains the approach to compartmentation as follows:

"The compartment annotation of new reactions was refined based on information from the UniProt and SGD databases. The subsystem annotation was firstly obtained from KEGG, and if no subsystems were found there, information from BioCyc or Reactome was used instead. If the reaction had no gene relations, we assumed that it occurred in the cytoplasm."

Minor comments:

Line 54: I'd be interested to see if the algorithm agrees with the literature on which genes are essential. If you remove the constraint that they can't be removed, are they kept anyway?

This is a complicated question due to the phenomenon of synthetic or bypass suppression, whereby the requirement for the activity of an essential gene is obviated by the loss of another gene's function. This has permitted the distinction between 'dispensable' and 'core' essential genes. We have added a new section [see p3, lines 93-105] to explain bypass suppression and show how effective our pipeline is at identifying core essential genes.

Line 65: Please provide references for the medium compositions here.

These are now given in Methods under 'Media Definition' [see pp 15-16, lines 501-528]

Line 69: I might have missed this, but is it known experimentally that yeast grows at the same rate in all conditions you tested? Did you optimize for the same doubling time in all conditions?

Yeast does not grow at the same rate in all conditions, but we optimised such that no more than the same fractional increase in growth rate (either 1% or 10%) was permitted for each condition.

Line 155: Wrong subfigure references, or subfigure labels are incorrect in the figure. Check the references. A few are missing the year.

This is now right. We have corrected the references/labels to the Figures. We have added the years in the References [see pp 12-13 and p19].

Reviewer #2 (Remarks to the Author):

The manuscript submitted by Jansen and colleagues reports the development of a computational pipeline to determine minimal metabolic networks (MMNs) with predicted growth outcomes above a certain threshold relative to the wild type. The pipeline starts with a metabolic model and a list of genes that are considered essential in the literature and cannot be deleted. An evolutionary algorithm is then used to iteratively delete genes and eventually reach the MMN. By working with the *S. cerevisiae* metabolic network in six different growth conditions, the authors have noticed a category of "mandatory" genes that are not essential but appear difficult to exclude from the MMN under most if or all conditions. More specifically, seven genes (named the magnificent seven by the authors) are always present in the MMN. The authors indicate that mandatory genes have more genetic and protein-protein interactions and a greater impact on the predicted growth than other genes taken at random in *S. cerevisiae*.

Overall, I found the topic of this manuscript to be interesting, and I believe that future work in synthetic biology will be aimed at building MMN. However, I have questions and concerns about the work that I think would be important to address:

I was disappointed that the predictions were only performed for one organism, *S. cerevisiae*. I appreciate the fact that six different conditions were evaluated but this limits the conclusions that can be drawn about mandatory genes. The authors mentioned that MMN for other organisms were also analyzed but that the results would be described elsewhere. This would have been a nice addition to understanding the general importance of mandatory genes, their conservation across species, and the type of reaction that they catalyze.

A new section demonstrating these interspecies comparisons and discussing their limitations has been added [see pp 7-9, lines 239-280 and new Figure 4].

I doubt that "mandatory" is the best term for this group of genes that are rarely omitted from MMN. Mandatory can bring confusion with essential, whereas in this case mandatory genes can be deleted individually (experimental data is demonstrating this) but not when metabolic reduction is brought to an extreme.

It is unclear after reading the manuscript how their conclusion is affected by the biomass objective function used by the model. The authors acknowledge this limitation and state that the cell biomass composition depends on old and potentially unreliable data. I wonder what the impact of modifying the biomass objective function would be, for example by using a computational approach that refines the exact components that the cell should make (PMID:

31009451).

Given that the list of mandatory genes could vary between species and potentially between conditions (for example if the biomass function varies), I am not convinced that the seven mandatory genes deserve to be called "magnificent". This leads the readers to think that these genes are special while they are not more important than strictly essential genes for cell growth. Finally, while I understand the difficulty of generating a cell with an MMN and that the authors refer to published experimental data (e.g. genetic or protein-protein interactions), having a few experiments trying to validate the difficulty of knocking-out combinations of mandatory genes (especially a few of the magnificent seven) would provide an important support for their conclusions. Otherwise, the conclusions remain purely based on computational predictions for which the validity remains hard to accurately estimate.

On a more minor note, I would like to tell the authors that Figure 1 can be a bit confusing since the pale blue that overlaps with the purple is very similar to the slightly darker blue used in the same panel. In addition, the Extended Data Table 1 is difficult to read because of the formatting of the last column on the right side of the page.

We prefer to keep these designations, which have been well received at international meetings of microbiologist, yeast geneticists, physiologists, and molecular biologists.

The definition of mandatory genes is articulated and is very specific, it must include the following 8 characteristics:

1) The mandatory genes are **not essential**, 2) they ensure both **viability and high growth rates**. 3) The mandatory genes are very **rarely eliminated in constructing a minimal metabolic network (MMN)**, suggesting that it is difficult for metabolism to be re-routed to obviate the need for such genes. 4) The removal of *mandatory* genes from the minimised metabolic network significantly **reduces its global efficiency**. 5) Bioinformatic analyses of the *mandatory* genes have revealed that not only do these genes have **more genetic interactions** than the bulk of metabolic genes 6) but their protein products also show **more protein-protein interactions**. 7) In yeast, *mandatory* genes are **predominantly single-copy** and 8) are **highly conserved across evolutionarily distant organisms**.

We have added a whole new section in which we show the results of experiments that aimed to construct some 25 multiple-deletion mutants including double, triple, and quadruple mutants and involving the Magnificent Seven genes, genes which are mandatory for growth under aerobic conditions, and a randomly chosen set of non-mandatory genes. The results fully support our conclusions [see pp 10-11, lines 314-350, Table1, and new Figure 5].

Reviewer #3 (Remarks to the Author):

Jansen et al. reported an in silico approach (based on an evolutionary algorithm) to reduce the number of genes in a given metabolic model without compromising biomass production. They applied it to a *Saccharomyces cerevisiae* metabolic model. The authors referred to the resulting metabolic network models as minimized metabolic networks (MMNs). The authors applied their approach to simulate effect of gene deletions in three different growth media in aerobic and anaerobic conditions. Remarkably, by analyzing the frequency of each metabolic gene in the resulting MMNs, authors proposed a new class of genes, mandatory genes, that is different from the essential gene category and seemed to be central for the proper flow of information in the resulting MMNs. Therefore, this work could contribute to our understanding of the general principles of metabolic networks while allowing us to re-evaluate the importance of potentially overlooked metabolic genes in *S. cerevisiae*. Although an appealing idea, there are several concerns that should be addressed by the authors in order to offer support to their claims (see comments below). In addition, the authors could gain additional insights by taking a closer look at the resulting MMNs.

Major comments

1. One of the main concerns with the study is the lack of model performance assessment and experimental validation. It may not be possible to validate predicted biomass of the MMNs (with

dozens of deleted genes), however authors should evaluate the accuracy of their metabolic network models to predict single gene deletions in the different growth contexts. They could achieve this by:

- Comparing the set of essential genes predicted by their metabolic models for each growth condition with available experimental data.
- Comparing biomass predictions for single gene KOs with data of growth effect of gene deletions.
- Comparing experimental growth of WT *S. cerevisiae* in the six growth conditions with the predicted biomass for WT.

We have added a whole new section in which we show the results of experiments that aimed to construct some 25 multiple-deletion mutants including double, triple, and quadruple mutants and involving the Magnificent Seven genes, genes which are mandatory for growth under aerobic conditions, and a randomly chosen set of non-mandatory genes. The results fully support our conclusions [see pp 10-11, lines 314-350, Table1, and new Figure 5].

2. Similarly, the proposed class of mandatory genes are based exclusively on computational predictions, no experimental support is given (not even for the magnificent seven genes). So, authors need to offer experimental support for this new gene category. To do so, authors could experimentally evaluate and compare the growth effect of single gene deletions of multiple mandatory and non-mandatory genes while comparing the biomass production with their model predictions.

See answer to 1., above.

3. The authors need to revise their list of essential genes in a context-specific manner, and re-run their analyses with the curated list of essential genes. During a quick literature

review of the magnificent seven genes, I found that several of them (e.g., TPS1, TPS2, CHO1, ADE3) have been classified as essential in minimal growth medium (see for example Heavner & Price 2015- cited by the authors, and other references such as PMID: 26426067, 19525417). Based on those previous reports, it is not surprising that deletions of most of those seven genes were predicted to completely inhibit growth.

We used the list of essential genes from the *Saccharomyces* Genome Database, which uses the standard definition of an essential gene for *S. cerevisiae*, see:

p3 lines 89-93:“The commonly used definition of essentiality for *S. cerevisiae* is that deletion of an essential gene results in failure to grow on a yeast extract/peptone/glucose (YPD) agar plate. These are extremely permissive conditions in which, for instance, auxotrophic or respiratory-deficient deletants are still able to grow”.

Thus, there will be other conditions in which the deletion of non-essential (minimal medium, a non-fermentable carbon source etc. The Reviewer is missing an important point, or perhaps re-stating it in their own way, that we make on p9, lines 291-295:

“gene essentiality is context dependent, that context including both the growth environment and the genetic background. Liu *et al.*³² conceive of essentiality as a quantitative trait that incorporates not only viability, but also evolvability. In this report, we show that mandatoriness is also a quantitative property that can be measured either as the proportion of *MMNs* that retain a given gene or in terms of a gene’s impact on the global efficiency of the metabolic network.”

As the authors stated in L58-l60, the definition of essentiality depends on the composition of the growth condition, therefore authors need to carefully review their list of essential genes taking

into account the conditions used in their simulations. Once authors have re- run their analyses, they can check if the new gene category they proposed is still present (see my comment # 2).

See response to comment # 3.

4. Given the immense space of possible combinations in the metabolic network (L478: 2^{976}), how reproducible are the results? If the authors run their simulations again, would they arrive to the same conclusions?

We ran each simulation experiment 3 times. We would note that these are not short simulations, each experiment required weeks of CPU time (in a computing farm with non-trivial computational resources in time and space). [see p. 2, lines 85-86]

5. Additional questions that the authors could address to get additional insights:

- Can the authors apply a different objective function (for example, ethanol production) and compare how different are the properties of the resulting MMNs and the corresponding mandatory genes?

Definitely yes, the designed framework can easily be applied by simply changing the objective function, for example the production of 1,4-butanediol or D-lactate or ethanol. In reality, we can do more, instead of changing the objective function we can *add more objective functions* according to the Pareto Optimal design paradigm. We can have as objective functions both biomass and the production of a particular chemical of interest. Combinations and possibilities offered by the framework are numerous and varied.

- How different are the fluxes of the reactions in the reduced networks with respect to the whole WT metabolic network?

As explained previously, there is no unique *MMN* for any of the conditions.

- Are there any emerging topological properties of the MMNs that distinguish them from other metabolic or biological networks?

In Figure 1, and in the analysis of the efficiency of minimal metabolic networks, we have shown that topological properties are less important than fluxes, the dynamics are determined by the weights in the arcs.

Moreover, there is an emerging topological property of the minimal metabolic networks:

The removal of *mandatory* genes from the minimised metabolic network significantly **reduces its global efficiency (property #4 of the mandatory genes)**.

Minor comments

6. Extended data table1

-What is the exact definition of MNs here? For example in row # 1, there are 787 MNs. Do you mean MMNs?

-In the same row #1, why do you use the nomenclature 81+4 mandatory genes instead of 85 genes?

It is written in the caption of the Table [p 1 Supplementary Information]. Maybe it wasn't clear.

"Models used in this study with the number of genes simulated in each of them and characteristics of Minimal Metabolic Networks found by our algorithm. A Default medium entry in the Medium column refers to a model that has not been changed from its original setting. The last column value is the number of mandatory genes divided in the number of genes always present in the MNs and the number of genes present in at least the 85% of the MNs but not in all of them. For *S. cerevisiae*

the last two columns show the shared mandatory genes in the aerobic or anaerobic results, or in both types.”

The last column value is the number of mandatory genes divided by the number of genes always present in the MMNs (100%) and the number of genes present in at least the 95% of the MNs but not in all of them.

7. L63-65: what was the point of generating an ad hoc minimal medium instead of using a standard minimal medium that would allow easier comparison with public data? Authors should consider replacing this condition with a standard minimal medium

Already did both.

8. L73: how many generations exactly? Please be specific.

5000 generations (as written in the Methods).

9. Figure 1

- Panel A: are the shown histograms for the 10% or 1% biomass threshold?

As stated in the text (P3, L90), we used the 1% threshold.

In Figure 1, panel A refer to the 1% biomass threshold. This information is included in the legend of the Figure [p 4, line 126].

- Panel B: It is unclear to me what the aerobic vs anaerobic represents. Specifically, how is different from the relevant comparison (e.g., top panel: SD aerobic and SD anaerobic)? Why is the yellow histogram different in each subpanel?

It is not – other than the fact that the different subpanels each deal with a different medium.

What is each condition compared to when defining the fraction of shared reactions?

In panel B, we report 3 figures.

In each of the 3 figures, we report the histograms

*) for two single media (the blue and orange histograms)

**) while with the yellow histograms, we report the fraction of reactions shared in the minimal metabolic networks considering both the aerobic and anaerobic conditions (we make the intersection of the reactions shared in the two media).

10. L102-103: do the authors mean higher number of transport-related genes in the SD aerobic model? It may help if authors highlight with a star or other symbol the transport-related categories in Extended Data Fig 3.

It is both aerobic and anaerobic.

Extended Data Fig 3 has been amended in the manner requested. [see p 9 Supplementary Information]

11. Extended Data Figure 2: It is unclear to me how the lyase activity category is related to the transport function. Can the authors explain this a little in the text?

The lyase category was chosen as a set of genes that were *not* transport-related and which therefore represent a comparator to the transport-related genes. If this is confusing, we can remove it.

12. L105-106: this sentence disagrees with Extended Data Figure 2a which shows a higher number of Kos in the SD anaerobic model with respect to the SD aerobic model.

It is true that Extended Data Figure 2a shows a higher number of KOs in the SD anaerobic model with respect to the SD aerobic model; in general, the Figure clearly shows that in the 3 anaerobic conditions there are more KOs than in the 3 aerobic conditions.

13. L121-122: It seems that the authors defined the mandatory gene set independently for aerobic and anaerobic conditions. If that's the case you need to mention it. Or do you define the mandatory genes for each condition?

We define mandatory genes for each condition. The Magnificent Seven are mandatory for ALL conditions examined.

14. L141: the concept of biomass pseudo-reaction could be explained in more detailed. The FVA abbreviation was introduced here without any background. It is also unclear why authors were interested in the pseudo-reaction and not the biomass itself.

This now done and an appropriate reference cited [see p. 15, lines 466-480].

The FVA abbreviation was introduced here without any background.

We are interested in FVA as it is very accurate to determine the maximum and minimum values of all the fluxes that will satisfy the constraints and allow for the same optimal objective value. Basically, FVA is a method to determine the range of possible reaction fluxes that still satisfy, within some optimality factor, the original FBA problem [see p. 15, lines 475-480].

It is also unclear why authors were interested in the pseudo-reaction and not the biomass itself.

Please see above.

15. Algorithms #2 and #5 and the functions that authors referred to in the methods section are briefly described in the text but there is no additional information in the supplement.

Done [see pages 16-18, lines 538-565m and pp.12-14 of Supplementary Information].

16. A supplement figure describing the algorithm used to derive the MMNs may be useful for the reader.

Done.

The algorithm used to derive the MMNs is the **Algorithm 2**. We added a sentence that reiterates this (under Algorithm 2, in the added description). [p 12 in the Supplementary Information].

REVIEWER COMMENTS

Reviewer #3 (Remarks to the Author):

Jansen et al. had addresses most of my concerns. However, a few remaining issues need to be addressed. see comments below:

1. L89-91: I now understand how the authors defined the set of essential genes for their work. However, the authors need to explicitly state the source of their essential genes here or in the methods. In the current version it is not obvious.
2. L100-L105: related to model predictions and experimental data comparison, authors stated that "Details of these analyses may be found in Extended Data" but I could not find this data.
3. L117-L120 and Extended Fig 2 (related to transport function genes): I do not agree with the authors' response. In the anaerobic context, the average number of KOs in the minimal medium seems to be smaller than in the SD medium. Is that difference statistically significant? Also, the authors should add to the figure legend the information about all functional categories used when defining the "transporter functional categories". For example, did the authors include ion transport, carbohydrate transport, etc.?
4. L305-306: overlap between gene essentiality predictions and experimental data from Leeuwen et al. should be included in the manuscript (at least in the supplement). Otherwise, author's claim is not supported. Additionally, the authors should evaluate not only the recall of their predictions but the precision. In other words, how many of the predicted essential genes were indeed reported as essential?
5. L309-313: same as my previous comment. Authors need to include the results of this comparison in the supplement.
6. Extended Data Table 1: despite authors reiteration of the figure legend, I think the last columns of the table are confusing. For example, in the second row, what does 117+2 mean? And the next value 46 + 3? 46 mandatory genes in which condition? 3 genes in which condition?
7. Related to my original comment # 5: authors may consider mentioning the possibility of multi-level optimization or different objective functions that may result in different MMNs. A minor point, but I disagree with the authors' response about their evaluation of MMNs topology. There are several topological properties that could be compared between the MMNs and the initial metabolic network (for example check Machicao et al. PMID: 30374088).
8. Extended Data Fig 2b: I would recommend that the authors remove this panel since it is redundant with Extended Fig 3. If they want to keep it, the lyase category should be removed.

Reviewer #3 (Remarks on code availability):

I could not find a README file in the repository. I did not try to run the code (it may require too much computational time due to the analyses performed).

Reviewer #4 (Remarks to the Author):

In this study, Jansen G et al. used a novel in silico synthetic biology pipeline to design minimal metabolic networks, eliminating the enzymes and transporters encoding genes. This resulted in a novel functional gene set they termed mandatory. The bioinformatics analysis further revealed that these mandatory genes have more genetic and protein-protein interactions than bulk metabolic genes and their products. While this is an interesting paper, I have several concerns that require additional supporting data to substantiate their claims. After reading the authors' responses to Reviewer 1's concerns, I believe they have not adequately addressed them. For instance, the authors did not comment on the difference between quasi-essential and mandatory genes, they did not provide any metabolic maps or create JSON or SBML files, and they did not clarify the ribosome compartment. In general, a lot of information is missing, and it is unclear how they generated the MMNs, making the study seem non-reproducible.

Previous concerns:

1. Previous reviewer 1 concern: I may have missed it but what is the doubling time for the wild type under all the various medium compositions? It seems you use a rule that any gene knockout producing a drop of 1- 10% in the growth rate of the wild type to identify the minimal metabolic networks, but how the mandatory genes are chosen is not clear. Under various medium conditions, I have seen the term quasi-essential used in the context of genes that are not essential but are needed to maintain a given doubling time within some reasonable range. Is this the same term as what you are calling "mandatory"?

Current reviewer comment: The authors are not consistent in their response. While the mandatory genes often exhibit these characteristics, this is not the same as quasi-essential genes.

2. Previous Reviewer 1 concern: Give the MMN as JSON or SBML files which would allow the use of Escher or some of software to present the metabolic network in Yeast 8.31 and then gray out the links that can be eliminated and still stay within the 1-10% grow rate cut-off.

Current reviewer comment: The requested file formats could be useful as additional resources. Regarding the proportion of MMNs, given the threshold built into the definition of a mandatory gene, designating those above 95% is less informative than knowing the exact score.

3. Based on authors response to previous reviewers' comments:

- I still have concerns about the nomenclature "mandatory genes"; this issue was also raised by one of the reviewers. After reviewing the authors' eight-point justification, I understand the difference between essential genes and mandatory genes, but I still do not see a clear justification for their choice of terminology.
- One of the reviewers raised concerns about using only one model organism, yeast. I did not find sufficient justification from the authors regarding this point. Additionally, they have not adequately addressed their criteria for selecting the specific genes they termed "mandatory."

Additional concerns from the current reviewer:

- Lines 38-42: The authors developed a computational pipeline, demonstrated using *S. cerevisiae*, and claimed it can be applied to all sequenced organisms. In my opinion, saying "any sequenced organisms" is not accurate. Instead, "a broad range of organisms" would be more appropriate, as not all organisms may be suitable for this analysis due to differences in metabolic complexity, data availability, or other factors.
- Lines 154-178: The Monte Carlo simulation was used to evaluate the impact of gene deletions on metabolic network efficiency. How efficient is this simulation in fully capturing the complexity of biological systems? The effect of gene deletions should be experimentally validated (e.g., gene knockdown experiments) to identify the functional importance of the identified genes.
- Lines 204-218: The study compared three characteristics (gene-gene interactions, protein-protein interactions, and paralogs) between mandatory, essential, and non-mandatory genes to evaluate functional importance in the metabolic network, finding significant differences. How did the authors rule out the impact of other influencing factors, such as gene expression levels, sub-cellular localization, or regulatory mechanisms, which could drastically affect network dynamics?
- Line 282: Please correct the figure number; it should be Figure 4.
- Lines 318-325, Table 1: The authors attempted to validate their pipeline predictions by constructing yeast strains with multiple deletions of either the Magnificent Seven genes or non-mandatory genes. While successful in constructing multiple deletion strains, the authors faced challenges in obtaining triple and quadruple deletions involving the Magnificent Seven genes compared to non-mandatory genes, reflecting inherent limitations in experimental approaches or

genetic interactions not fully captured by computational models. How do the authors explain this limitation of their pipeline?

- Lines 650-652, Method: For the in vivo gene deletion experiments, non-mandatory genes were selected randomly, which could lead to the exclusion of functionally important genes and inclusion of less important ones. Would it be possible to complement the gene selection strategy by prioritizing genes based on functional annotations and/or computational predictions of gene essentiality?
- Lines 660-666, Method: How was the experiment on solid media performed and analyzed? Were the experiments and analysis performed in a blinded manner?

REPLIES TO REVIEWER COMMENTS

Please note that all line numbers cited below refer to the manuscript read by the Reviewers and NOT to the current version of the paper.

Reviewer #3 (Remarks to the Author):

Jansen et al. had addresses most of my concerns. However, a few remaining issues need to be addressed. see comments below:

1. L89-91: I now understand how the authors defined the set of essential genes for their work. However, the authors need to explicitly state the source of their essential genes here or in the methods. In the current version it is not obvious.

In the last version of the manuscript, we cited the *Saccharomyces* Genome Database (ref. 12) as the source of the list of essential genes. For the avoidance of doubt, we now name the database and give its URL in the text. In addition, a full list of the essential genes that our algorithm was prohibited from removing is provided in Supplementary Information Table 1.

2. L100-L105: related to model predictions and experimental data comparison, authors stated that “Details of these analyses may be found in Extended Data” but I could not find this data.

We apologise to the Reviewer. This information was omitted in error and has now been included in Supplementary Data Table 4, which is now referred to in the text.

3. L117-L120 and Extended Fig 2 (related to transport function genes): I do not agree with the authors’ response. In the anaerobic context, the average number of KOs in the minimal medium seems to be smaller than in the SD medium. Is that difference statistically significant?

The legend to this Figure has been amended appropriately.

Also, the authors should add to the figure legend the information about all functional categories used when defining the “transporter functional categories”. For example, did the authors include ion transport, carbohydrate transport, etc.?

A new Supplementary Table (ST5 transporters_related_genes.xlsx) ... has been added to Supplementary Information that lists all the transporter genes included in the WT metabolic mode; it also provides information on their functional categories (amino acid transport, ammonium transport, ion transport etc.) as well as the SGD description of their functional role (SGD accessed on 22.06.24).

4. L305-306: overlap between gene essentiality predictions and experimental data from Leeuwen et al. should be included in the manuscript (at least in the supplement). Otherwise, author’s claim is not supported. Additionally, the authors should evaluate not only the recall of their predictions but the precision. In other words, how many of the predicted essential genes were indeed reported as essential?

See response to point 2

5. L309-313: same as my previous comment. Authors need to include the results of this comparison in the supplement.

See response to point 2.

6. Extended Data Table 1: despite authors reiteration of the figure legend, I think the last columns of the table are confusing. For example, in the second row, what does 117+2 mean? And the next value 46 + 3? 46 mandatory genes in which condition? 3 genes in which condition?

We agree that this was confusing and have removed the +X designation (which was intended to provide a count of genes just below the threshold) from the Table.

7. Related to my original comment # 5: authors may consider mentioning the possibility of multi-level optimization or different objective functions that may result in different MMNs.

This is a good point, and the required comment has now been included in Methods.

A minor point, but I disagree with the authors' response about their evaluation of MMNs topology. There are several topological properties that could be compared between the MMNs and the initial metabolic network (for example check Machicao et al. PMID: 30374088).

We thank the Reviewer for referring us to Machicao *et al.* We had included such an analysis in an early draft of our Ms, but later expunged it since we felt it added little of substance. Please see below for this excised material:

Figure 1 Numerical and network properties of the *S.cerevisiae* MNs in 7 different conditions and for 2 Growth Rate threshold. (a) Frequencies of MNs active genes; two different clusters are present, for aerobic and anaerobic conditions. (b) Fraction frequency of MNs shared active reactions through a pairwise comparison. Considering the networks in the same conditions the shared reactions are 60~90%, while comparing networks in different conditions the fraction is 30~50%. (c) For each condition the MNs active genes number using a Growth rate threshold of 1% or 10% is reported. On average less genes are required for the less strict bound. (d) Sum of reactions fluxes producing one

of two energy molecules (ATP, NADH), highlighting different regions in aerobic and anaerobic conditions and for the two thresholds. (e) Average weight distribution in MNs; aerobic and anaerobic networks have different distributions. (f) Average degree distribution in MNs, similar for all the conditions MNs. (g) Plot and frequency distribution of the mean and standard deviation of weights in the MNs, divided in three operating regions

In the body of the text, we reported:

We have also used basic measures from the complex network theory²⁵ to describe the MNs (Fig. 1e-1g) considered as a bipartite graph of reactions and metabolites; there are two different behaviours in the weights distributions for aerobic and anaerobic (a pattern that is maintained from the Wild Type networks), while the degree distributions are all similar. The mean and standard deviation of the weights (Fig. 1g) show how the aerobic networks in different conditions are divided in two sub-clusters.

We now include parts e-g, with an appropriate comment, in the Supplementary Information (Figures 10-12, pp. 24-26). These data, and the Figure, are now referred to in the text as part of the legend to Fig. 2.

8. Extended Data Fig 2b: I would recommend that the authors remove this panel since it is redundant with Extended Fig 3. If they want to keep it, the lyase category should be removed.

We agree and have now removed the lyase category as being too vague a functional class to be useful. Now we only have Figure 2a (as new Figure 2).

Reviewer #3 (Remarks on code availability):

I could not find a README file in the repository. I did not try to run the code (it may require too much computational time due to the analyses performed).

We have updated and cleaned up the code on GitHub to make it easier to follow; we have also ensured that that the README file is available. It follows the new URL (included in the manuscript):

https://github.com/GiuseppeNicosia1/MinimalNetwork_CompleteCode

Reviewer #4 (Remarks to the Author):

In this study, Jansen G et al. used a novel in silico synthetic biology pipeline to design minimal metabolic networks, eliminating the enzymes and transporters encoding genes. This resulted in a novel functional gene set they termed mandatory. The bioinformatics analysis further revealed that these mandatory genes have more genetic and protein-protein interactions than bulk metabolic genes and their products. While this is an interesting paper, I have several concerns that require additional supporting data to substantiate their claims. After reading the authors' responses to Reviewer 1's concerns, I believe they have not adequately addressed them. For instance, the authors did not comment on the difference between quasi-essential and mandatory genes,

We now see the potential for confusion between “quasi-essential” and “mandatory”. The main point is that the impact of deleting a *mandatory* gene is that it has a quantitative impact that can be measured

either as the proportion of *MMNs* that retain a given gene or in terms of a gene's impact on the global efficiency of the metabolic network. Accordingly, we have abandoned the term “mandatory” and now refer to the genes as Network Efficiency Determinants (*NEDs*).

they did not provide any metabolic maps or create JSON or SBML files,

The JSON files are now available at

<https://drive.google.com/file/d/1SEMdTbwx5rh60Sa1tddBGDeTSCRTlu1W/view?usp=sharing>

Five metabolic maps, constructed using different constraints or nutrient environments, may be found as Figures 5-9 (pp.19-23) of the Supplementary Information.

and they did not clarify the ribosome compartment.

The data for compartments, as for functional categories, were obtained from the GO Slim terms representing major biological processes, molecular functions, and cellular components; they are then summarized and used to categorize the genes in the model and generate the boxplots.

In general, a lot of information is missing, and it is unclear how they generated the *MMNs*, making the study seem non-reproducible.

We do not understand the Reviewer's comment. We provided a general description of the pipeline (*L43-L54* and *L68-L86*) and details of our algorithm were provided in Methods (*L530-L576*). Moreover, we provide access to the code, which has now been cleaned to improve its clarity and is available in GitHub.

Previous concerns:

1. Previous reviewer 1 concern: I may have missed it but what is the doubling time for the wild type under all the various medium compositions?

This has been added to Extended Data Table 1.

It seems you use a rule that any gene knockout producing a drop of 1- 10% in the growth rate of the wild type to identify the minimal metabolic networks, but how the mandatory genes are chosen is not clear. Under various medium conditions, I have seen the term quasi-essential used in the context of genes that are not essential but are needed to maintain a given doubling time within some reasonable range. Is this the same term as what you are calling “mandatory”?

Current reviewer comment: The authors are not consistent in their response. While the mandatory genes often exhibit these characteristics, this is not the same as quasi-essential genes.

On 1137-1139, we stated:

A striking feature revealed by these analyses is that there is a set of genes that are present in >95% of all *MMNs*, when using the 1% threshold for a reduction in growth rate.

The text in red, above, has been added to the new version in order to provide further clarity. Moreover, as explained above, we have now abandoned the term *mandatory* in favour of Network Efficiency Determinant (*NED*).

2. Previous Reviewer 1 concern: Give the MMN as JSON or SBML files which would allow the use of Escher or some of software to present the metabolic network in Yeast 8.31 and then gray out the links that can be eliminated and still stay within the 1-10% grow rate cut-off.

Current reviewer comment: The requested file formats could be useful as additional resources. Regarding the proportion of MMNs, given the threshold built into the definition of a mandatory gene, designating those above 95% is less informative than knowing the exact score.

A link to the JSON files

<https://drive.google.com/file/d/1SEMdTbwx5rh60Sa1tddBGDeTSCRTlu1W/view?usp=sharing> is now provided with the legend to Extended Data Fig. 8. Figures 8-12 (pp.16-200 of the Supplementary Information gives 5 metabolic maps constructed (using Escher) under different constraints or nutrient environments.

Based on authors response to previous reviewers' comments:

- I still have concerns about the nomenclature "mandatory genes"; this issue was also raised by one of the reviewers. After reviewing the authors' eight-point justification, I understand the difference between essential genes and mandatory genes, but I still do not see a clear justification for their choice of terminology

Fully answered above, the term Network Efficiency Determinant (*NED*) has been introduced.

- One of the reviewers raised concerns about using only one model organism, yeast. I did not find sufficient justification from the authors regarding this point. Additionally, they have not adequately addressed their criteria for selecting the specific genes they termed "mandatory."

It is simply not true that we have confined ourselves to a single species, *Sacharomyces cerevisiae*. We have constructed *MMNs* for 31 other species and this exercise is extensively discussed, L243-L295 and Fig. 4. We make no apology for concentrating many of our efforts on *S. cerevisiae* since not only is the metabolic model of this organism the most up to date and extensively tested, but there is also a full suite of ancillary data (e.g. a comprehensive genome-wide experimental study of gene-gene interactions etc.). For these reasons, and its ease of genetic manipulation, we chose *S. cerevisiae* to make the multiple knock-outs necessary for the validation of our predictions. Whilst this might be done be done with *Schizosaccharomyces pombe*, we have no experience of laboratory work with the fission yeast. Moreover, the ancillary data available for *Sz. pombe* is far less extensive than that available for *S. cerevisiae*.

Additional concerns from the current reviewer:

- Lines 38-42: The authors developed a computational pipeline, demonstrated using *S. cerevisiae*, and claimed it can be applied to all sequenced organisms. In my opinion, saying "any sequenced organisms" is not accurate. Instead, "a broad range of organisms" would be more appropriate, as not

all organisms may be suitable for this analysis due to differences in metabolic complexity, data availability, or other factors.

We thank the Reviewer for making this very important point. The requested change has been made.

- Lines 154-178: The Monte Carlo simulation was used to evaluate the impact of gene deletions on metabolic network efficiency. How efficient is this simulation in fully capturing the complexity of biological systems?

We did not carry out a brute force approach as it is infeasible due to the combinatorial nature of the problem which makes it computationally intractable. We used Monte Carlo simulations which are quite computationally efficient and produce accurate and reliable results. We used a number of trials = 10^4 .

The effect of gene deletions should be experimentally validated (e.g., gene knockdown experiments) to identify the functional importance of the identified genes.

This has been done (see below).

- Lines 204-218: The study compared three characteristics (gene-gene interactions, protein-protein interactions, and paralogs) between mandatory, essential, and non-mandatory genes to evaluate functional importance in the metabolic network, finding significant differences. How did the authors rule out the impact of other influencing factors, such as gene expression levels, sub-cellular localization, or regulatory mechanisms, which could drastically affect network dynamics?

All of these factors could indeed affect network dynamics but were not included in the original metabolic model and so could not account for the results generated by our pipeline. We have confined these comparisons to cases where there is a direct relation to gene *function* (gene-gene and protein-protein interactions), or gene retention (paralogy) and for which comprehensive, genome-wide data are available for *S. cerevisiae*. Gene *regulation* does not come into this category.

Line 282: Please correct the figure number; it should be Figure 4.

We thank the Reviewer for pointing this out. The correction has been made.

- Lines 318-325, Table 1: The authors attempted to validate their pipeline predictions by constructing yeast strains with multiple deletions of either the Magnificent Seven genes or non-mandatory genes. While successful in constructing multiple deletion strains, the authors faced challenges in obtaining triple and quadruple deletions involving the Magnificent Seven genes compared to non-mandatory genes, reflecting inherent limitations in experimental approaches or genetic interactions not fully captured by computational models. How do the authors explain this limitation of their pipeline?

We apologise to the Reviewer for allowing brevity to override clarity in our description of these experiments and their interpretation. We did not face ‘challenges’ in obtaining triple or quadruple mutants of Magnificent Seven genes, rather the inability to obtain such mutants demonstrates that there are synthetic lethal (or negative epistatic) interactions between the Magnificent Seven genes and that is why the *MMNs* predicted by our pipeline simulations never contain multiple deletions of

these genes. This is because we always had a positive selection each successive deletion event, as explained in Methods (L655-L658):

Multi-gene deletants were constructed by deleting genes of interest sequentially from single-gene deletion mutants (carrying *kanMX*) via multiple rounds of target gene replacement using either antifungal resistance cassettes (*natNT2*, *hphMX6*) or a *URA3* marker⁴⁰. Successful transformants were confirmed via colony PCR. All 657 primers used for construction of the deletion cassettes and confirmatory PCRs are listed in Supplementary Table 6 (ST6).

The successful isolation of all mutants attempted among the non-*NED* genes demonstrates that our operational ability to perform gene deletion is robust. So, if no viable mutants are obtained with some combinations of Magnificent Seven genes, this indicates that the mutants are either lethal or so sick that are unable to recover after transformation. Hence the triple and quadruple mutant combinations of the Magnificent Seven genes are lethal precisely because there are genetic interactions between them, and that is why they are retained in all the *MMNs*. In other words, our modelling and simulation studies have revealed these higher-order interactions and they have been confirmed by *in vivo* experiments.

We have now re-written this section of our manuscript and trust that its clarity has been improved.

- Lines 650-652, Method: For the *in vivo* gene deletion experiments, non-mandatory genes were selected randomly, which could lead to the exclusion of functionally important genes and inclusion of less important ones. Would it be possible to complement the gene selection strategy by prioritizing genes based on functional annotations and/or computational predictions of gene essentiality?

Cherry picking of specific genes would allow preconceptions to bias the experimental design. According to the model we used, all the non-*NED* genes are in hierarchically equivalent so it is not possible to prioritise genes.

However, ***functionally important genes were not excluded by our random-picking methodology.*** Specifically, one non-*NED* gene, *HTD2*, which encodes a mitochondrial 3-hydroxyacyl-thioester dehydratase, was in the pool of randomly selected genes for the *in vivo* gene deletion experiments. *HTD2* is crucial for respiration and, consequently, its deletion impacts colony size and growth rate. Even in this case, the two non-*NED* triple mutants with *htd2Δ* (*chs1Δ/dnf3Δ/htd2Δ* and *suc2Δ/chs1Δ/htd2Δ*) exhibited significantly better growth than the Magnificent Seven triple mutant (*ynk1Δ/gpt2Δ/tps2Δ*) in both YPD and SD media (see Fig. 5), supporting the claim that the efficiency of the metabolic network is more dependent on the Magnificent Seven genes.

- Lines 660-666, Method: How was the experiment on solid media performed and analyzed? Were the experiments and analysis performed in a blinded manner?

The way the solid growth experiments were performed was fully explained, both in the legend to the Figure presenting the results (L332-L337):

Figure 5. Growth characteristics of multiply deletant strains. (A) Spot test assay of: Magnificent Seven gene 332 triple mutant (*ynk1Δ/_gpt2Δ/_tps2Δ*), non-mandatory gene triple mutants (*chs1Δ/dnf3Δ/htd2Δ*, *chs1Δ/dnf3Δ/suc2Δ*, *suc2Δ/chs1Δ/htd2Δ*, *suc2Δ/chs1Δ/dnf3Δ*), and BY4741 WT on (i) YPD; and (ii) SD.

and in the Methods (I663-I667):

For solid growth assays, overnight cultures were harvested and diluted with water to reach a final OD₆₀₀=4. Ten-fold serial dilutions of the cell suspensions, with concentrations ranging from OD₆₀₀=0.4 to OD₆₀₀=4x10⁻⁶, were spotted onto the agar plates (YPD and SD) and photographed after 48h incubation.

The solid media experiment is a spot assay (Fig. 5A), which involves ten-fold serial dilutions of a liquid culture and the “spotting” of each dilution on solid media. The experiment semi-quantitatively investigates yeast growth (*i.e.* number/density of colonies) on solid media plates. This is a straightforward, commonly used, and widely accepted method within the experimental community (*i.e.* never referenced).

It is not clear to us why the Reviewer asks about “blinding”. All the cultures were spotted at the same time, and several technical replicates were conducted on different plates to ensure consistency. Blinding is usually used when there is some subjective element that can prejudice the set-up of the experiment or the assessment of the data; neither is true here. Moreover, uncropped photographs of the drop-out plates are provided in Figure 4 of the Supplementary Methods in SI as an assurance of the probity of these data. In addition, we would point out that the growth curves for the liquid cultures and their exponential growth rates were generated automatically using a plate reader. Supplementary Table 7 presents growth rate, doubling times, and yields in both SD and YPD for the wild-type strains and the multiple mutants that we constructed.

REVIEWERS' COMMENTS

Reviewer #3 (Remarks to the Author):

Jansen et al. had addressed most of my concerns. See below my comments regarding some minor issues that remain:

1. Authors should add a legend tab to each of their supplementary files so they can explain the content of each tab. For example, the complete tab in the supp file 4, why some genes do not have a value for the presence in MNNs column? It's unclear.
2. Additionally, I was not able to find the list of genes that were considered as essential and prohibited from removing by the algorithm in supp file 1.
3. Related to the comparison with the core genes defined by Leeuwen et al.: how many of the genes labelled as core by Leeuwen et al. were among the genes considered as essential by the algorithm? In other words, for the 36 genes labelled as indispensable by Leeuwen et al. that were not deleted in the MNN, was that because their deletion was prohibited or actually the algorithm decided to not delete them?
4. Related to the comparison with the analysis by Warringet et al.: I still could not find the comparison between the MNNs and the data by Warringer et al.

Reviewer #3 (Remarks on code availability):

I checked the README file. I did not try to run the code.

Reviewer #4 (Remarks to the Author):

After reviewing the authors' responses to the comments, I believe they have appropriately revised the manuscript. Some reviewer suggestions, such as adding more organisms or including gene localization, are beyond the scope of this paper. The GitHub link includes a README file, and the JSON files are accessible. I am satisfied with the rebuttal and find the manuscript to be an interesting and valuable resource that should be accepted for publication.

However, there are a few minor errors in the text still. Despite my previous request for the authors to update the terms "quasi-essential" and "mandatory" genes, Figure 2 still uses the original nomenclature of mandatory and non-mandatory genes. Additional minor corrections are needed:

- Line 134 should correct "using allowing" to simply "using."
- Figure 3, Table 1 title, and lines 439, 440, and 441, as well as all methods sections, use different fonts from the rest of the paper.
- Line 370 contains an incomplete sentence.
- Line 375 should be corrected from "so sick that cannot" to "so sick that they cannot."

Aside from these points, I have no further corrections to suggest for the authors.

RESPONSE TO REVIEWERS' COMMENTS

Reviewer #3 (Remarks to the Author):

Jansen et al. had addressed most of my concerns. See below my comments regarding some minor issues that remain:

1. Authors should add a legend tab to each of their supplementary files so they can explain the content of each tab. For example, the complete tab in the supp file 4, why some genes do not have a value for the presence in MNNs column? It's unclear.

Legend tabs now included (Supplementary Table 0)

2. Additionally, I was not able to find the list of genes that were considered as essential and prohibited from removing by the algorithm in supp file 1.

In ST1, for each of the two Tabs for *S. cerevisiae*, *S.cerevisiae* (1% GR thr.) and *S.cerevisiae* (10% GR thr.) starting from row number 978 there are the essential genes. All the genes are sorted in descending order by frequency. At the end there are the essential genes that the algorithm never turns off. This point is reiterated in the legend to ST1 in ST0 and the essential genes are given in **bold** in ST1.

3. Related to the comparison with the core genes defined by Leeuwen et al.: how many of the genes labelled as core by Leeuwen et al. were among the genes considered as essential by the algorithm? In other words, for the 36 genes labelled as indispensable by Leeuwen et al. that were not deleted in the MNN, was that because their deletion was prohibited or actually the algorithm decided to not delete them?

This is all made clear in ST4 and its legend (in ST0). Both have been vetted and approved by Prof Boone (Toronto).

4. Related to the comparison with the analysis by Warringet et al.: I still could not find the comparison between the MNNs and the data by Warringer et al.

We now expand on this comparison in the text. As we state, the differences in growth rate, although statistically significant, are (necessarily) very small. Therefore, we do not consider it is appropriate to add an extra Fig. There was no significant difference for either the length of the lag phase or the biomass yield.

Reviewer #3 (Remarks on code availability):

I checked the README file. I did not try to run the code.

Fine

Reviewer #4 (Remarks to the Author):

After reviewing the authors' responses to the comments, I believe they have

appropriately revised the manuscript. Some reviewer suggestions, such as adding more organisms or including gene localization, are beyond the scope of this paper. The GitHub link includes a README file, and the JSON files are accessible. I am satisfied with the rebuttal and find the manuscript to be an interesting and valuable resource that should be accepted for publication.

Thank you!

However, there are a few minor errors in the text still. Despite my previous request for the authors to update the terms "quasi-essential" and "mandatory" genes, Figure 2 still uses the original nomenclature of mandatory and non-mandatory genes. Additional minor corrections are needed:

- Line 134 should correct "using allowing" to simply "using."

Done

- Figure 3, Table 1 title, and lines 439, 440, and 441, as well as all methods sections, use different fonts from the rest of the paper.

Re-formatting the Ms to comply with *Nature Comm* rules (rather than *Nature*) has fixed most of these problems. In any case, the journal will employ its own fonts in preparing the proofs.

- Line 370 contains an incomplete sentence.

Rectified

- Line 375 should be corrected from "so sick that cannot" to "so sick that they cannot."

Done.